# NDACC harmonized formaldehyde time-series from 21 FTIR stations covering a wide range of column abundances

Corinne Vigouroux[1], Carlos Augusto Bauer Aquino[2], Maite Bauwens[1], Cornelis Becker[3], Thomas Blumenstock[4], Martine De Mazière[1], Omaira García[5], Michel Grutter[6], César Guarin[6], James Hannigan[7], Frank Hase[4], Nicholas Jones[8], Rigel Kivi[9], Dmitry Koshelev[10], Bavo Langerock[1], Erik Lutsch[11], Maria Makarova[12], Jean-Marc Metzger[13], Jean-François Müller[1], Justus Notholt[14], Ivan Ortega[7], Mathias Palm[15], Clare Paton-Walsh[8], Anatoly Poberovskii[12], Markus Rettinger[15], John Robinson[16], Dan Smale[16], Trissevgeni Stavrakou[1], Wolfgang Stremme[6], Kim Strong[11], Ralf Sussmann[15], Yao Té[10], and Geoffrey Toon[17]

[1]Royal Belgian Institute for Space Aeronomy (BIRA-IASB), Brussels, Belgium
[2]Instituto Federal de Educaçao (IFRO), Ciência e Tecnologia de Rondônia, Porto Velho, Brazil
[3]Stichting Atmospherische en Hydrologische Ontwikkeling (SAHO), Paramaribo, Suriname
[4]Karlsruhe Institute of Technology (KIT), IMK-ASF, Karlsruhe, Germany
[5]Izaña Atmospheric Research Centre (IARC), Agencia Estatal de Meteorología (AEMET), Santa Cruz de Tenerife, Spain
[6]Centro de Ciencias de la Atmósfera, Universidad Nacional Autónoma de México (UNAM), 04510 Mexico City, México
[7]Atmospheric Chemistry, Observations & Modeling, National Center for Atmospheric Research (NCAR), Boulder, CO, USA
[8]Centre for Atmospheric Chemistry, University of Wollongong, Wollongong, Australia
[9]Finnish Meteorological Institute (FMI), Sodankylä, Finland
[10]LERMA-IPSL, Sorbonne Université, CNRS, PSL Research University, Observatoire de Paris, 75005 Paris, France
[11]Department of Physics, University of Toronto, Toronto, Canada
[12]Saint Petersburg State University, Atmospheric Physics Department, St. Petersburg, Russia
[13]Observatoire des Sciences de l'Univers Réunion (OSU-R), UMS 3365, Université de la Réunion, Saint-Denis, France
[14]Institute of Environmental Physics, University of Bremen, Bremen, Germany
[15]Karlsruhe Institute of Technology (KIT), IMK-IFU, Garmisch-Partenkirchen, Germany
[16]National Institute of Water and Atmospheric Research Ltd (NIWA), Lauder, New Zealand
[17]Jet Propulsion Laboratory, California Institute of Technology, Pasadena, California, USA

*Correspondence to:* C. Vigouroux
(Corinne.Vigouroux@aeronomie.be)

**Abstract.** Among the more than twenty ground-based FTIR (Fourier Transform infrared) stations currently operating around the globe, only a few have provided formaldehyde (HCHO) total columns time-series until now. Although several independent studies have shown that the FTIR measurements can provide formaldehyde total columns with a good precision, the spatial coverage has not been optimal for providing good diagnostics for satellite or model validation. Furthermore, these past studies used different retrieval settings, and biases as large as 50% can be observed in the HCHO total columns depending on these retrieval choices, which is also a weakness for validation studies combining data from different ground-based stations.

For the present work, the HCHO retrieval settings have been optimized based on experience gained from the past studies and have been applied consistently at the 21 participating stations. Most of them are either part of the Network for the Detection of Atmospheric Composition Change (NDACC), or under consideration for membership. We provide the harmonized settings and a characterization of the HCHO FTIR products. Depending on the station, the total systematic and random uncertainties of an

individual HCHO total column measurement lie between 12 and 27%, and between 1 and $11\times10^{14}$ molec/cm$^2$, respectively. The median values among all stations are 13% and $2.9\times10^{14}$ molec/cm$^2$, for the total systematic and random uncertainties, respectively.

This unprecedented harmonized formaldehyde data set from 21 ground-based FTIR stations is presented and its comparison to
a global chemistry transport model shows its consistency, in absolute values as well as in seasonal cycles. The network covers very different concentration levels of formaldehyde, from very clean levels at the limit of detection (few $10^{13}$ molec/cm$^2$) to highly polluted levels ($7\times10^{16}$ molec/cm$^2$). Because the measurements can be made at any time during daylight, the diurnal cycle can be observed and is found to be significant at many stations. These HCHO time-series, some of them starting in the 1990's, are crucial for past and present satellite validation, and will be extended in the coming years for the next generation of
satellite missions.

## 1  Introduction

Through reactions with hydroxyl radical (OH) and NOx (NO+NO$_2$), the volatile organic compounds (VOCs) exert a strong influence on the oxidizing capacity of the atmosphere. These reactions produce ozone and secondary organic aerosols, which
affect air quality and global climate. Given their short lifetimes (from a few minutes to a few hours for the most reactive ones, Kesselmeier and Staudt (1999)) and their different sources depending on geographical locations, it is very difficult to derive a global atmospheric burden for most of the VOCs from current measurements. The observation of formaldehyde (HCHO), which is an intermediate product of the degradation of many non-methane VOCs (NMVOCs) and has a lifetime of only a few hours, allows to constrain the NMVOCs emissions and to test our understanding of the complex and still uncertain degradation mech-
anisms of these NMVOCs (Stavrakou et al., 2009). The use of satellite HCHO measurements in combination with tropospheric chemistry transport models to derive NMVOCs emissions has been the subject of several past studies (e.g. Palmer et al. (2003); Millet et al. (2008); Stavrakou et al. (2009); Fortems-Cheiney et al. (2012); Barkley et al. (2013); Marais et al. (2014)). The past and present HCHO satellite data sets include those from GOME (1996-2003), SCIAMACHY (2003-2012), OMI (2004-), GOME2A (2006-), OMPS (2011-), GOME2B (2012-), and very recently TROPOMI (2017-). The NMVOCs emissions de-
rived from the top-down approaches using these satellite data sets rely on the accuracy of the measurements. An indirect way to test these accuracies is to compare the emission budgets obtained using two different satellite data sets as in Barkley et al. (2013) for SCIAMACHY and OMI, or in Stavrakou et al. (2015) for OMI and GOME2. While the global emission budgets are in general consistent (Stavrakou et al., 2015), there are large differences on the top-down estimates on a regional scale, e.g. differences up to nearly 50% are observed over Amazonia between SCIAMACHY and OMI (Barkley et al., 2013), and up to
nearly 25% between GOME2 and OMI (Stavrakou et al., 2015). To conclude unambiguously whether these differences are due to biases in the satellite products (due to retrieval settings, vertical sensitivities, horizontal resolution,...) or to the diurnal cycle of formaldehyde (SCIAMACHY and GOME-2 measuring in the morning and OMI in the afternoon) requires validation with independent and accurate ground-based measurements (Barkley et al., 2013; De Smedt et al., 2015; Stavrakou et al., 2015).

At present, validation studies of HCHO satellite products have been performed at a few locations only, mainly using aircraft data (Martin et al., 2004; Barkley et al., 2013; Zhu et al., 2016), the MAX-DOAS (Multi-AXis Differential Optical Absorption Spectroscopy) technique (Wittrock et al., 2006; De Smedt et al., 2015) and the FTIR (Fourier Transform Infra-Red) technique (Jones et al., 2009; Vigouroux et al., 2009; De Smedt et al., 2015). This is not sufficient to provide a good picture of the satellites' accuracy, especially given the high geographical variability of formaldehyde. A lot of effort is therefore currently underway to increase the number of ground-based stations providing HCHO data, using the DOAS or the FTIR technique, initiated in view of the TROPOMI Cal/Val activities. This paper presents the work accomplished in this direction using FTIR measurements at most of the NDACC (Network for the Detection of Atmospheric Composition Change) stations, and including some more recent observing stations, that will also be part of the NDACC in the near future.

Up to now, time-series of HCHO total columns have been studied at six FTIR stations only, among the more than 20 FTIR sites currently in operation: Ny-Alesund (Notholt et al., 1997), Wollongong (Paton-Walsh et al., 2005), Lauder (Jones et al., 2009), Reunion Island (Vigouroux et al., 2009), Eureka (Viatte et al., 2014), and Jungfraujoch (Franco et al., 2015). We note that HCHO has also been measured by the JPL MkIV instrument (Toon, 1991) at various ground-based sites since 1985 (see http://mark4sun.jpl.nasa.gov/ground.html), although these data are not used in this work due to their very different acquisition, and analysis procedures. The main reasons for having so few FTIR HCHO data available are: 1) that it is challenging to find robust retrieval settings for this species that has weak absorption signatures in the infrared, which are in addition surrounded by strong lines from interfering gases; 2) that HCHO is not part of the NDACC FTIR target species (which are $O_3$, $HNO_3$, HF, HCl, CO, $CH_4$, $N_2O$, $ClONO_2$, HCN, and $C_2H_6$, publicly available at http://www.ndsc.ncep.noaa.gov/clickmap/). In the above cited studies, different retrieval settings are used, although the retrieved HCHO total columns can be very sensitive to some of them: e.g. a positive bias of 30% or even 50% is found at Reunion Island if the settings of Franco et al. (2015) or Jones et al. (2009) are used, respectively, instead of those from Vigouroux et al. (2009). Although these high biases are consistent with the uncertainty budgets, it is important, to facilitate the interpretation of a satellite or model validation, to harmonize the settings among the stations. Therefore, in the present work, we have set up common retrieval settings that can be used at any ground-based site, even under very humid conditions or low HCHO concentrations. These settings will be described in Sect. 2 together with a characterization of the retrieved HCHO products, i.e., their averaging kernels and uncertainty budget. The complete time-series of HCHO total columns obtained at the 21 participating stations are shown in Sect. 3, as well as the diurnal cycles and a short assessment of the long-term trends. We then use the chemistry-transport model IMAGES (Stavrakou et al., 2015), which provides data for the 2003-2016 period, to show the consistency of our harmonized FTIR data sets: comparisons between FTIR and IMAGES monthly mean time-series and seasonal cycle at the 21 stations are presented in Sect. 4.

## 2   Ground-based FTIR HCHO data: description and characterization

### 2.1   FTIR HCHO monitoring

Table 1 lists the ground-based FTIR stations included in this study, while Fig. 1 shows their geographical distribution. These stations perform regular solar absorption measurements, under clear-sky conditions, using either the high-resolution spec-

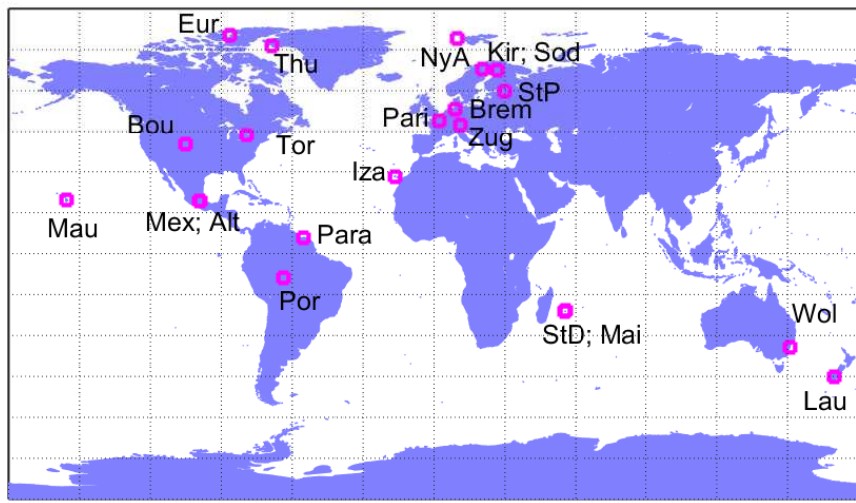

**Figure 1.** Location of the FTIR stations providing HCHO total columns.

trometers Bruker 120 M, 125 M, 120 HR, and/or 125 HR which can achieve a spectral resolution of 0.0035 cm$^{-1}$ or better, or the Bomem DA8 which can achieve a spectral resolution of 0.004 cm$^{-1}$. The only lower spectral resolution spectrometer (0.06 cm$^{-1}$) used in this study is the Bruker Vertex at Mexico City. This instrument is not accepted by the NDACC FTIR standards at present, therefore Mexico City is the only site in this study that will not be part of NDACC.

5     The formaldehyde spectral signatures used in ground-based infrared measurements lie in the 3.6 $\mu$m region and belong to the $\nu_1$ and $\nu_5$ bands. This implies that for HCHO, a CaF$_2$ or KBr beamsplitter and a nitrogen-cooled InSb detector are used, together with an optical filter which usually covers the 2400-3310 cm$^{-1}$ region (so-called NDSC-3 filter, see e.g. Senten et al. (2008)). At St. Petersburg a broader filter is used (1700-3400 cm$^{-1}$). The spectral resolution can be reduced in order to increase the signal to noise ratio (SNR). In practice, the spectra used in the present study have a resolution between 0.0035 and 0.009

10   cm$^{-1}$, except for Mexico city (0.075 cm$^{-1}$).

    HBr or N$_2$O cell measurements are regularly performed to verify the alignment of the instruments. The instrument line shape (ILS) can be obtained by analyzing these cell measurements using the LINEFIT program (Hase et al., 1999). This ILS can impact the shape of gas absorption lines, and its determination by LINEFIT can be used as an input parameter in the forward model of the retrieval codes (Sect. 2.2).

15  **2.2   Harmonized retrieval strategy**

We refer to e.g. Pougatchev et al. (1995) and/or Hase et al. (2004) for more details on the FTIR retrieval principles. Total columns of atmospheric gases, but also volume mixing ratio vertical profiles are obtained from their pressure- and temperature-dependent absorption lines. As seen in Table 1, two retrieval algorithms are used in the NDACC FTIR community: PROFITT9 (Hase et al., 2006), and SFIT2 (Pougatchev et al., 1995) which has been updated to SFIT4 09.4.4. It has been demonstrated

**Table 1.** Characteristics of the FTIR stations contributing to the present work: location and altitude (in km a.s.l.), time-period used in the present study, instrument type, retrieval code, team.

| Station | Latitude | Longitude | Altitude | Time-period | Instrument | Code | Team |
|---|---|---|---|---|---|---|---|
| Eureka | 80.05° N | 86.42° W | 0.61 | 2006–2016 | Bruker 125 HR | SFIT4 | U. of Toronto |
| Ny-Alesund | 78.92° N | 11.92° E | 0.02 | 1993–2017 | Bruker 120/5 HR | SFIT4 | U. of Bremen |
| Thule | 76.52° N | 68.77° W | 0.22 | 1999–2016 | Bruker 120 M | SFIT4 | NCAR |
| Kiruna | 67.84° N | 20.40° E | 0.42 | 2005–2016 | Bruker 120/5 HR | PROFFIT | KIT / IMK–ASF |
| Sodankyla | 67.37° N | 26.63° E | 0.19 | 2012–2017 | Bruker 125 HR | SFIT4 | FMI & BIRA |
| St. Petersburg | 59.88° N | 29.83° E | 0.02 | 2009–2017 | Bruker 125 HR | SFIT4 | SPb State U. |
| Bremen | 53.10° N | 8.85° E | 0.03 | 2004–2017 | Bruker 125 HR | SFIT4 | U. of Bremen |
| Paris | 48.97° N | 2.37° E | 0.06 | 2011–2016 | Bruker 125 HR | PROFFIT | LERMA |
| Zugspitze | 47.42° N | 10.98° E | 2.96 | 1995–2017 | Bruker 120/5 HR | PROFFIT | KIT / IMK–IFU |
| Toronto | 43.60° N | 79.36° W | 0.17 | 2002-2016 | Bomem DA8 | SFIT4 | U. of Toronto |
| Boulder | 40.04° N | 105.24° W | 1.61 | 2010-2016 | Bruker 120 HR | SFIT4 | NCAR |
| Izaña | 28.30° N | 16.48° W | 2.37 | 2005–2016 | Bruker 125 HR | | |
| Mauna Loa | 19.54° N | 155.57° W | 3.40 | 1995–2016 | Bruker 120/5 M | SFIT4 | NCAR |
| Mexico City | 19.33° N | 99.18° W | 2.26 | 2013–2016 | Bruker Vertex 80 | PROFFIT | UNAM |
| Altzomoni | 19.12° N | 98.66° W | 3.98 | 2012–2016 | Bruker 120/5 HR | PROFFIT | UNAM |
| Paramaribo | 5.81° N | 55.21° W | 0.03 | 2004–2016 | Bruker 120/5 M | SFIT4 | U. of Bremen |
| Porto Velho | 8.77° S | 63.87° W | 0.09 | 2016–2017 | Bruker 125 M | SFIT4 | BIRA |
| Saint-Denis | 20.90° S | 55.48° E | 0.08 | 2004–2011 | Bruker 120 M | SFIT4 | BIRA |
| | | | | 2011-2013 | Bruker 125HR | | |
| Maïdo | 21.08° S | 55.38° E | 2.16 | 2013–2017 | Bruker 125 HR | SFIT4 | BIRA |
| Wollongong | 34.41° S | 150.88° E | 0.03 | 1996–2007 | Bomem DA8 | SFIT4 | U. of Wollongong |
| | | | | 2007–2016 | Bruker 125 HR | | |
| Lauder | 45.04° S | 169.68° E | 0.37 | 2001–2016 | Bruker 120 HR | SFIT4 | NIWA |

in Hase et al. (2004) that the profiles and total column amounts retrieved from these two different algorithms under identical conditions are in excellent agreement.

We summarize in Table 2 the forward model and retrieval parameters that have been harmonized. The forward model uses pressure and temperature profiles from NCEP (National Centers for Environmental Prediction) for each site, except that the temporal resolution can vary depending on the retrieval team from daily means, 6-hourly ones, or even hourly interpolated ones.

The dominant source of systematic uncertainty being the spectroscopic parameters, it is crucial that all stations use the same spectroscopic database. We use the compilation from G. Toon (JPL), the so-called atm16 linelist, which is available

**Table 2.** Summary of the HCHO harmonized forward model and retrieval parameters. The micro-windows limits are given in $cm^{-1}$.

| | |
|---|---|
| Pressure and temperature profiles | NCEP |
| Spectroscopic database | atm16 (=HITRAN 2012 for HCHO) |
| Solar lines | SFIT4.09.4.4 |
| Micro-windows | MW #1: 2763.42 - 2764.17 |
| | MW #2: 2765.65 - 2766.01 |
| | MW #3: 2778.15 - 2779.1 |
| | MW #4: 2780.65 - 2782.0 |
| Deweighted spectral sections | 2780.967 - 2780.993 ($O_3$) |
| | 2781.42 - 2781.48 ($CH_4$) |
| Retrieved species | HCHO, HDO, $CH_4$, $O_3$, $N_2O$, solar lines |
| | optional: $CO_2$, $H_2O$ |
| *a priori* profiles (except HDO and $H_2O$) | WACCM v4 |
| Regularization | Tikhonov L1 |

at http://mark4sun.jpl.nasa.gov/toon/linelist/linelist.html. In this atm16 linelist, the HCHO and $N_2O$ lines correspond to the HITRAN 2012 database (Rothman et al., 2013). This HITRAN 2012 database includes the latest improved HCHO parameters (broadening coefficients, Jacquemart et al. (2010)), which complements the release in HITRAN 2008 (Rothman et al., 2009) of new line intensities from the same group (Perrin et al., 2010). The water vapor and its isotopologues lines in atm16 are

5   from Toth 2003 [1]; some lines of the other strong absorbing gases in the vicinity of HCHO ($O_3$ and $CH_4$) have been empirically adjusted or replaced with older HITRAN versions in atm16 when obvious problems were found in the HITRAN 2012 database. For the CO solar lines, we use the linelist updated from Hase et al. (2010) that is distributed in the NDACC community (SFIT4 package v09.4.4).

To avoid any bias between the stations due to different spectroscopic parameters, it is also mandatory to harmonize the
10   spectral micro-windows (MW) containing the HCHO signatures. The challenge of the HCHO retrievals is that this species

[1]http://mark4sun.jpl.nasa.gov/data/spec/H2O/RAToth_H2O.tar

has very weak absorption signatures in the infrared (below 1%), and it is therefore very important to minimize the impact of the interfering gases having more intense signatures, either by avoiding micro-windows with strong interfering lines when feasible, or by including them only in case they are very well fitted (e.g. no large residuals remain due to bad spectroscopic or incorrect ILS parameters). In the past studies, while the micro-window spectral widths differ, some common HCHO sig-
natures were used: the two more intense signatures at about 2778.5 cm$^{-1}$ and 2781.0 cm$^{-1}$ were used in all previous studies (Notholt et al., 1997; Paton-Walsh et al., 2005; Viatte et al., 2014; Jones et al., 2009; Vigouroux et al., 2009), except in Franco et al. (2015) who discarded the 2781.0 cm$^{-1}$ signature because of the bad residuals due to poorly fitted CH$_4$ lines (from HITRAN 2008, Rothman et al. (2009)). In Vigouroux et al. (2009), in which HITRAN 2004 (Rothman et al., 2005) was used, the micro-windows containing these two stronger signatures were quite narrow (2778.20 - 2778.59; 2780.80 - 2781.15 cm$^{-1}$),
in order to minimize residuals due to neighboring CH$_4$ lines. With the empirically improved CH$_4$ spectroscopy in atm16, we can use larger windows (see Table 2 and Fig. 2), with the advantages of fixing more the background and the interfering species, leading to an improved precision and accuracy in the HCHO total columns. We keep the two narrow micro-windows used in Vigouroux et al. (2009) and Franco et al. (2015) at about 2763.5 and 2765.8 cm$^{-1}$, which contain less absorption from interfering gases, but the gain in information, the so-called degrees of freedom for signal (DOFS, see Rodgers (2000)), is relatively
small (0.1-0.2, compared to about 1.0 to 1.5 from the two main windows).

We give in Fig. 2 an example of a spectrum calculated from the retrieval using a spectrum recorded on the 12-02-2014 at Maïdo and corresponding to a retrieved HCHO total column of 2.48×10$^{15}$ molec/cm$^2$, a DOFS of 1.1, and a root-mean-square (RMS) of 0.11, which compares well to the mean obtained for all measurements at Maïdo of 2.00×10$^{15}$ molec/cm$^2$, 1.2, and 0.12, for columns, DOFS and RMS, respectively. The corresponding residuals (calculated - observed spectra) are shown in
Fig. 3, when the spectroscopic parameters are taken from HITRAN 2012 and with the atm16 empirical linelist. We can see the improvement in MW #1 obtained simply by changing the line position of an O$_3$ line (2763.8598 cm$^{-1}$ instead of 2763.8588 cm$^{-1}$). The spectroscopic parameters in MW #2 are the same in both cases, the little improvement seen in this MW is due to the better fitting of the other MWs, that allows better calculated profiles for all gases. The CH$_4$ line in MW #3 is poorly fitted using the HITRAN 2012 linelist, and the improvement in the atm16 is due to a change in several spectroscopic parameters
(line position, line intensity,...). The two more intense CH$_4$ lines in MW #4 have also been improved by using the atm16 linelist. However, to further improve the fits, one CH$_4$ line and one O$_3$ line were empirically deweighted (see Table 2). The comparison of these two linelists shows the crucial need for good spectroscopic parameters in order to obtain precise amounts of atmospheric gases. As seen in Fig. 3 (right panel), the residuals are not perfect and there is still room for further improvement in forward model parameters. The atm linelist created by G. Toon (JPL) is updated each four years when HITRAN provides a
new release, so that when the HITRAN linelist is improved and provides either similar or better residuals than the atm linelist, the empirical parameters of atm are changed by the preferred official database.

In SFIT4 and PROFFIT retrieval codes, based on optimal estimation, *a priori* information (profile and regularization matrix) needs to be provided. In this work, the *a priori* HCHO profile, as well as all interfering species except water vapor and its isotopologues, were provided for each station from the v4 of the model WACCM (Garcia et al., 2007). A single profile for
each species is used in the time-series retrievals and corresponds to the mean of the model profiles calculated at each station

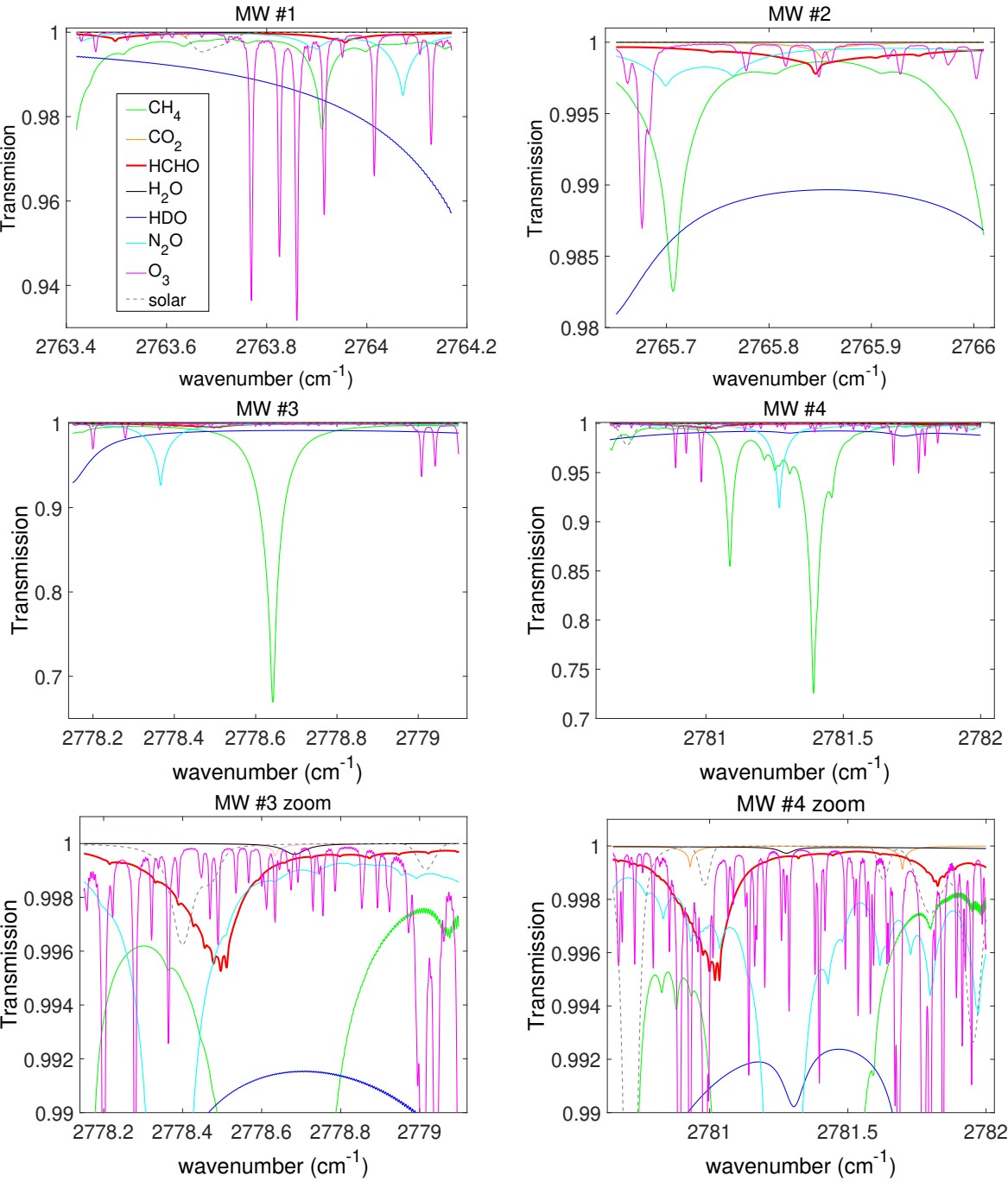

**Figure 2.** Retrieved contributions of all fitted species in the four MWs (upper and middle panels) used in the analysis for a spectrum recorded on 12-02-2014 at Maïdo and corresponding to a retrieved HCHO total column of $2.48 \times 10^{15}$ molec/cm$^2$. The lower panels are magnifications of the MWs #3 and #4.

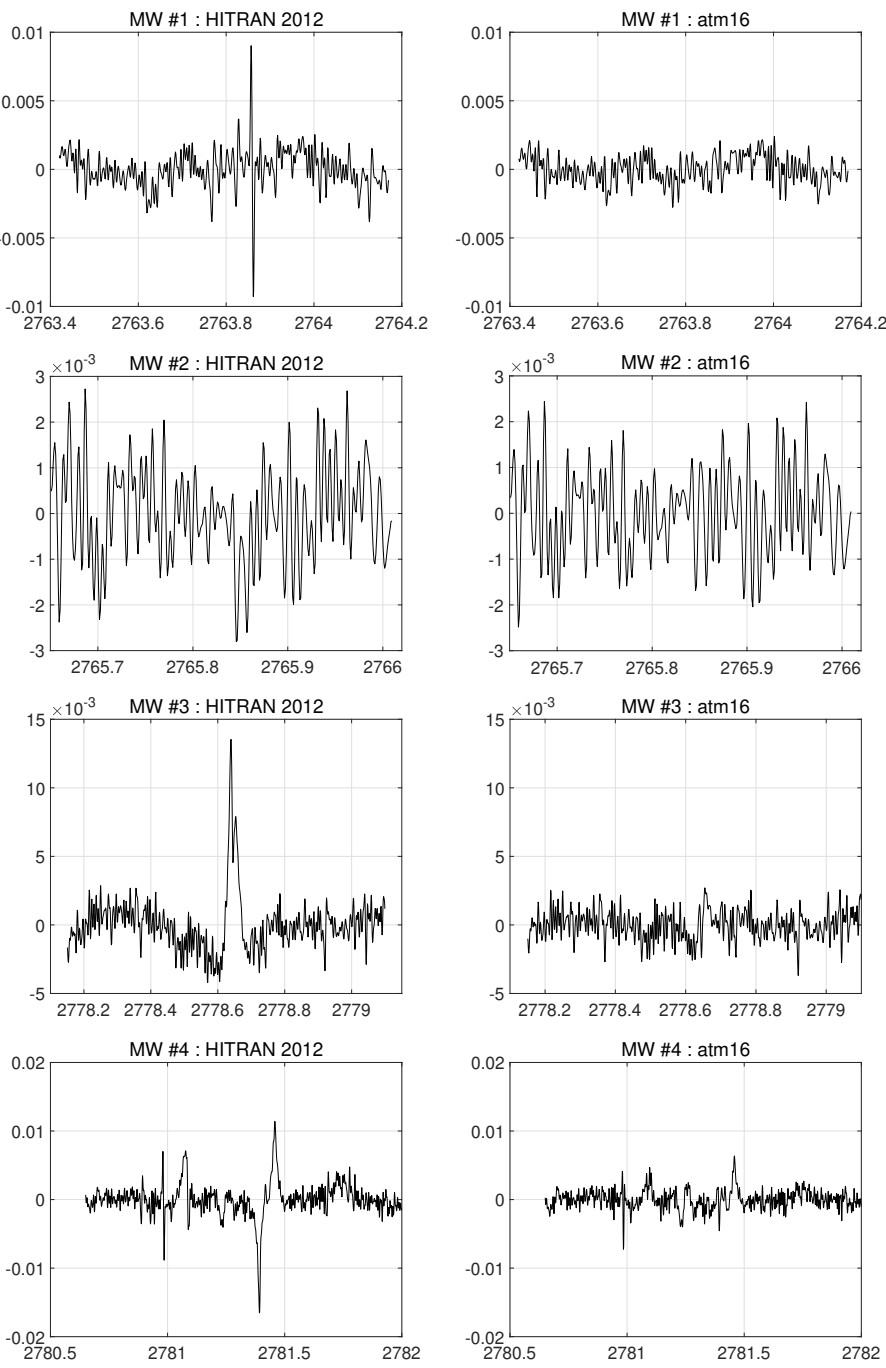

**Figure 3.** Residuals (calculated - observed spectrum) in each of the four MWs for the retrieval of a spectrum recorded on 12-02-2014 at Maïdo and corresponding to a retrieved HCHO total column of $2.48 \times 10^{15}$ molec/cm$^2$. The x-axis represents the wavenumber in cm$^{-1}$. The left panels are obtained when the HITRAN 2012 spectroscopy is used, and the right panels show the improvement made by using the atm16 linelist.

from 1980 to 2020. For $H_2O$ and HDO, which have a high atmospheric variability, it is usually preferred (except at the stations Lauder, Mexico City and Altzomoni) not to use a single *a priori* profile: for each individual spectrum, the water vapor *a priori* profiles are taken either from the 6-hourly vertical profiles provided by NCEP, or from independent preliminary profile retrievals. The $H_2O$ absorption being very weak in the chosen MWs, and the HDO profile being retrieved simultaneously with

HCHO, the impact of using a single *a priori* profile at the three cited stations is assumed to be small. For the regularization matrix $\mathbf{R}$, we followed Vigouroux et al. (2009) and Sussmann et al. (2011) and used *ad hoc* Tikhonov (Tikhonov, 1963) L1 regularization as described e.g. in Steck (2002), for the reason that we do not have realistic *a priori* covariance matrix $\mathbf{S}_{\mathrm{var}}$ from other measurements sources, especially with a good vertical resolution. The regularization matrix $\mathbf{R} = \alpha \mathbf{L_1}^T \mathbf{L_1}$ is used in most cases for the determination of HCHO low vertical resolution profiles, but also for profile retrievals of the interfering

species when improvement is observed compared to the fit of a single scaling factor (which is applied to the *a priori* profiles). This is the case for HDO and $CH_4$ for which profile retrievals are made, and at some stations for $O_3$. For the stations Kiruna, Izaña, Zugspitze, and Paris, a scaling of HCHO *a priori* profiles is preferred to a Tikhonov regularization, but due to the low DOFS available for this species (see Sect. 2.3), this has little influence on the retrieved total columns (below 2% when tested at Maïdo). For the other stations, the $\alpha$ values are site dependent, since it can depend e.g. on the HCHO amounts or the SNR

of the spectra. Note that the SNR value can be the "real" one coming from the inherent noise in each spectrum, but also can be chosen as an "effective" SNR, that is used as well as a regularization parameter. This effective SNR is smaller than the real one, since the residuals in a spectral fit are not only coming from pure measurement noise but also from uncertainties in the forward model parameters. The regularization choice ($\alpha$ and SNR if an effective one is used) is made at each station in order to obtain stable retrievals (no "overfitting") with significant decrease of the residuals (no "underfitting"), as in the well-known

L-curve method (Hansen , 1992).

It is worth noting that another important forward model parameter is the instrumental line shape (ILS) since it impacts the gases absorption line shapes. The treatment of ILS in the retrievals has not been harmonized yet among the stations because the stability and quality of the alignment is site dependent and/or the instrument's PIs have their own preferences. This is however another step toward full harmonization that should be done in the future within NDACC. At present, there are three options

for considering the ILS and we refer to Vigouroux et al. (2015) for more details. In the present work, the NIWA, NCAR and University of Bremen stations use a constant and ideal ILS (both modulation efficiency and phase error), i.e. the spectrometers are perfectly aligned. This is a valid approximation based upon LINEFIT ILS analysis of HBr cell spectra measurements (Sect. 2.1). The IMK-ASF, LERMA and UNAM stations use fixed ILS parameters that are previously retrieved using the cell measurements and the LINEFIT code (Hase et al., 1999). For the other stations, the effective apodization parameter is retrieved

simultaneously with the target species profiles, while the phase error parameter is assumed to be ideal.

## 2.3 Characterization: averaging kernels and uncertainty budget

The vertical resolution and sensitivity of the retrieved HCHO products can be characterized by the averaging kernel matrix $\mathbf{A}$ (Rodgers, 2000):

$$\mathbf{A} = (\mathbf{K}^T \mathbf{S}_\epsilon^{-1} \mathbf{K} + \mathbf{R})^{-1} \mathbf{K}^T \mathbf{S}_\epsilon^{-1} \mathbf{K}, \tag{1}$$

where $\mathbf{K}$ is the weighting function matrix that links the measurement vector $\boldsymbol{y}$ to the state vector $\boldsymbol{x}$: $\boldsymbol{y} = \mathbf{K}\boldsymbol{x} + \boldsymbol{\epsilon}$, with $\boldsymbol{\epsilon}$ representing the measurement error. In our retrievals, we assume $\mathbf{S}_\epsilon$ to be diagonal, with the diagonal elements being the inverse square of the SNR. $\mathbf{R}$ is the regularization matrix which, in this work, has been chosen as the Tikhonov L1 matrix (see Sect. 2.2).

We give the trace of this averaging kernel matrix $\mathbf{A}$ for the elements corresponding to the HCHO profiles, the so-called DOFS, in Table 3, for each station. The DOFS are ranging from 1.0 to 1.5, meaning that we can not provide more than one piece of information on the vertical profile. This is the reason why only total columns of HCHO are discussed in this paper, and not vertical profiles. We show in Fig. 4 (upper panels) the averaging kernels (AK, rows of $\mathbf{A}$), for four different stations, having DOFS ranging from 1 (only scaling) to 1.5. Similar averaging kernels are obtained for the other stations with similar DOFS (not shown). We can observe that, in each case, the AK peak at about the same altitude (8 km) with full-width-at-half-maximum of about 16-18 km, showing that we have limited vertical resolution, and that we are sensitive to the whole troposphere mainly, and to a lesser extent to the lowermost stratosphere. The total column averaging kernel (TotAK), to be associated with the FTIR retrieved total columns, is plotted as well. The associated *a priori* profiles are also shown in Fig. 4 (lower panels) for completeness, together with the mean and standard deviation of the retrieved profiles. As expected by the low DOFS, the shape of the retrieved profiles is very similar to the shape of the *a priori* profiles.

The uncertainty budget is calculated following the formalism of Rodgers (2000), and can be divided into three different sources: the measurement noise uncertainty (purely random), the forward model parameter uncertainties (random and systematic), and the smoothing error expressing the uncertainty due to the limited vertical resolution of the retrieval (random and systematic). At each station, the random uncertainty (square root of sum of squares of the measurement noise error and of all the random forward model errors) and the systematic uncertainty (square root sum of the squares of all systematic errors) are calculated for each single measurement. Except for a few cases (NCAR stations and Wollongong) for which a typical smoothing error is given, and St. Petersburg for which the mean value for 2013 is given, the smoothing uncertainty is also calculated for each individual measurement. We give in Table 3 the mean of the random and systematic uncertainties, of the smoothing uncertainties (both random and systematic parts), and of the total random/systematic uncertainties (square root sum of the squares of the random/systematic error and the smoothing random/systematic error), obtained using the FTIR complete time-series, at each station.

The random uncertainty given in Table 3 is dominated at all sites by the measurement noise whose error covariance matrix $\mathbf{S}_n$ is calculated as:

$$\mathbf{S}_n = \mathbf{G}_y \mathbf{S}_\epsilon \mathbf{G}_y^{\mathbf{T}}, \tag{2}$$

**Table 3.** Mean of the HCHO total columns (TC), in $10^{14}$ molec/cm$^2$, and Degrees of Freedom for Signal (DOFS) obtained at each FTIR station. The stations with strictly 1 DOFS (Kiruna, Izaña, Zugzpitze, and Paris) only make a scaling of the HCHO *a priori* profile, i.e. no change in the vertical shape of the *a priori* profile is allowed. We give, in $10^{14}$ molec/cm$^2$, the mean of 1) the random uncertainties (Rand) that were calculated for each individual HCHO total column (excluding the smoothing part); 2) the smoothing random error (Smoo Rand); 3) the total random error (Total Rand=$\sqrt{\text{Rand}^2 + \text{Smoo Rand}^2}$). We also provide the total random error in % for completeness. We give the mean of the systematic uncertainties in %: first without the smoothing part (Syst), then the smoothing systematic error (Smoo Syst), and the total systematic error (Total Syst=$\sqrt{\text{Syst}^2 + \text{Smoo Syst}^2}$). If Rodgers and Connor (2003) methodology is used in model/instrument comparisons, only the Rand and Syst uncertainties need to be taken into account (not the total errors). We provide in addition the mean differences between two subsequent FTIR measurements taken within 30 minutes, in both absolute ($10^{14}$ molec/cm$^2$) and percent units (Diff30), relative to mean TC. The PROFFIT stations are indicated with (***).

| Station | DOFS | mean TC | Rand | Smoo Rand | Total Rand | Syst | Smoo Syst | Total Syst | Diff30 |
|---|---|---|---|---|---|---|---|---|---|
| Eureka | 1.3 | 12.7 | 1.0 | 0.6 | 1.2 (9.3%) | 12.2% | 3.5% | 12.8% | 1.5 (11.7%) |
| Ny-Alesund | 1.6 | 15.8 | 1.8 | 0.5 | 1.9 (11.7%) | 13.3% | 3.4% | 13.8% | 3.9 (24.9%) |
| Thule | 1.1 | 15.7 | 1.3 | 0.9 | 1.5 (9.8%) | 14.3% | 3.8% | 14.8% | 1.8 (11.7%) |
| Kiruna*** | 1 | 17.5 | 3.5 | 0.8 | 3.6 (20.8%) | 25.6% | 8.6% | 27.1% | 0.7 (3.8%) |
| Sodankyla | 1.1 | 25.4 | 1.5 | 1.7 | 2.3 (9.0%) | 13.4% | 3.8% | 14.1% | 2.4 (9.3%) |
| St. Petersburg | 1.4 | 59.4 | 2.6 | 2.1 | 3.3 (5.6%) | 13.9% | 2.4% | 14.2% | 2.8 (4.6%) |
| Bremen | 1.2 | 59.6 | 2.3 | 1.7 | 2.9 (4.8%) | 12.9% | 2.9% | 13.3% | 3.1 (5.2%) |
| Paris*** | 1 | 73.0 | 5.3 | 1.4 | 5.5 (7.6%) | 16.3% | 4.6% | 17.0% | 3.3 (4.8%) |
| Zugspitze*** | 1 | 12.3 | 2.2 | 0.5 | 2.3 (18.6%) | 20.7% | 5.8% | 21.7% | 1.0 (8.0%) |
| Toronto | 1.3 | 95.1 | 5.1 | 4.1 | 6.7 (7.1%) | 12.6% | 2.7% | 13.0% | 19.3 (20.4%) |
| Boulder | 1.1 | 57.6 | 2.6 | 3.9 | 4.7 (8.2%) | 12.7% | 2.1% | 13.0% | 5.3 (9.2%) |
| Izaña*** | 1 | 20.4 | 3.3 | 0.2 | 3.3 (16.0%) | 20.9% | 4.4% | 21.4% | 0.8 (4.0%) |
| Mauna Loa | 1.1 | 10.1 | 1.4 | 1.0 | 1.8 (17.3%) | 12.5% | 3.8% | 13.1% | 1.4 (14.0%) |
| Mexico City*** | 1.0 | 220.9 | 11.1 | 2.5 | 11.4 (5.2%) | 12.0% | 1.2% | 12.1% | 24.0 (10.9%) |
| Altzomoni*** | 1.1 | 21.8 | 2.3 | 1.2 | 2.6 (11.7%) | 16.0% | 3.2% | 16.3% | 2.3 (10.5%) |
| Paramaribo | 1.5 | 64.3 | 3.4 | 1.3 | 3.6 (5.6%) | 12.2% | 3.1% | 12.7% | 11.9 (18.5%) |
| Porto Velho | 1.1 | 190.0 | 3.5 | 8.3 | 9.1 (4.8%) | 12.8% | 4.1% | 13.5% | 5.9 (3.1%) |
| Saint-Denis | 1.2 | 38.8 | 2.2 | 0.8 | 2.4 (6.1%) | 13.4% | 4.3% | 14.1% | 2.8 (7.2%) |
| Maïdo | 1.2 | 20.0 | 1.4 | 0.4 | 1.4 (7.3%) | 12.9% | 2.3% | 13.1% | 1.1 (5.6%) |
| Wollongong | 1.5 | 78.9 | 3.0 | 2.2 | 3.7 (4.7%) | 10.9% | 3.0% | 11.6% | 11.6 (15.0%) |
| Lauder | 1.4 | 25.6 | 1.5 | 0.4 | 1.6 (6.3%) | 12.4% | 2.6% | 12.8% | 3.6 (14.0%) |
| Median | 1.1 | 25.6 | 2.3 | 1.2 | 2.9 (7.6%) | 12.9% | 3.4% | 13.5% | 2.8 (9.3%) |

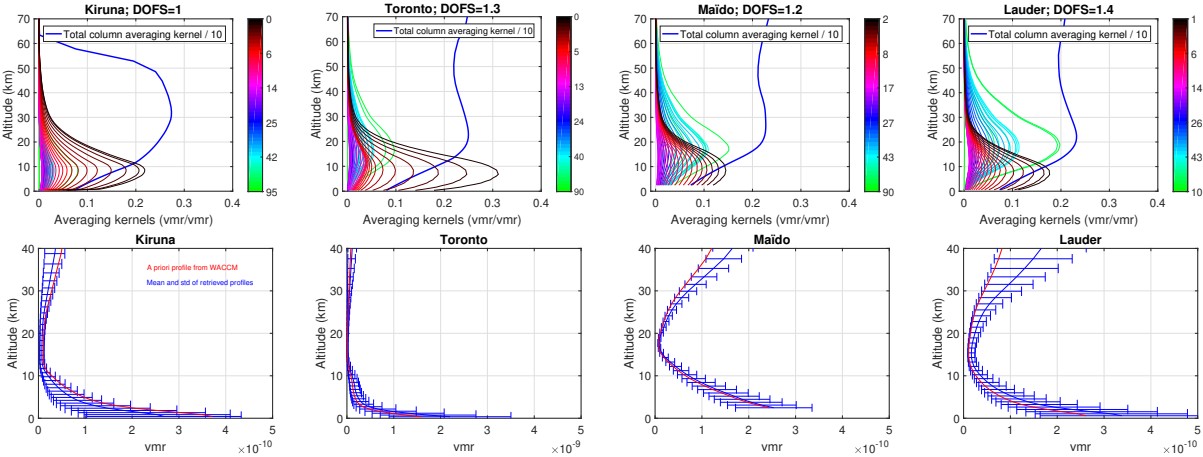

**Figure 4.** Upper panels: averaging kernels (rows of $\mathbf{A}$) and total column averaging kernel for four of the FTIR stations, with DOFS ranging from 1.0 to 1.4. The total column averaging kernel is also shown in thick blue line (divided by 10 for visibility). The color code for the different averaging kernels depending on their altitude is given in the color bar in km. Lower panels: *a priori* profiles from the WACCM v4 model (red), and the mean and standard deviation of the retrieved profiles, for the same four stations.

where $\mathbf{S}_\epsilon$ is assumed to be diagonal, with the square of the inverse of the SNR as diagonal elements, and $\mathbf{G}_y$ denotes the contribution matrix $\mathbf{A} = \mathbf{G}_y\mathbf{K}$. In this calculation of the measurement noise error, the SNR must be the real one coming from the noise in the spectra, and not a regularization one as can be chosen in the retrieval process (as in Eq. 1; see also Sect. 2.2). For the HCHO spectra used in this study, this SNR can vary between 100 for the worst cases and 3000, with a mean of about 700-1000 for the Bruker 120/5 HR instruments, and 500 for the Bomem DA8.

The forward model parameters error covariance matrices $\mathbf{S}_f$ are calculated according to:

$$\mathbf{S}_f = (\mathbf{G}_y\mathbf{K}_b)\mathbf{S}_b(\mathbf{G}_y\mathbf{K}_b)^{\mathbf{T}}, \tag{3}$$

where $\mathbf{S}_b$ is the covariance matrix of $\mathbf{b}$, the vector of forward model parameters. For each individual forward model parameter, the $\mathbf{K}_b$ sensitivity matrix is mostly calculated by using analytic derivatives, while the covariance matrix $\mathbf{S}_b$ is an estimate of the uncertainty on the model parameter itself.

Effort has been made in this study to harmonize the uncertainty budget at all sites. This is done by calculating across the network the errors from the same forward model parameters (solar zenith angle, temperature, spectroscopic line parameters, baseline, ...) and by choosing the same $\mathbf{S}_b$ matrix for relevant parameters (i.e. when they are not site or instrument dependent like e.g. for the spectroscopic line parameters). However, some differences remain between the SFIT4 and PROFFIT codes that result in small differences still occurring between the two groups of users, despite the use of harmonized parameters. For the SFIT4 users, the random uncertainty given in Table 3 is dominated by the measurement noise (Eq. 2). We see from Table 3 that the random error is between 1.0 and $3.6 \times 10^{14}$ molec/cm$^2$ for the SFIT4 stations equipped with the high-resolution Bruker spectrometers 120/5 HR or M (the higher values coming from the 120/125 M instruments at Paramaribo and PortoVelho),

while it can reach $5.1 \times 10^{14}$ molec/cm$^2$ with the Bomem DA8 in Toronto. For the PROFFIT users, the random uncertainty is calculated a little bit larger (from 3.5 to 5.3 $\times 10^{14}$ molec/cm$^2$) for the sites with high-resolution spectrometers, and $11.1 \times 10^{14}$ molec/cm$^2$ with the low-resolution spectrometer Bruker Vertex 80 at Mexico City. The main difference between SFIT4 and PROFFIT is the additional error calculated at the PROFFIT stations due to the channeling of the spectra. However, we give also in Table 3 the mean differences between two subsequent FTIR measurements taken within 30 minutes (Diff30), as an upper limit for the total random uncertainty: this difference can be larger than the error budget if HCHO has faster variability than 30 minutes, but with enough statistics, the mean differences should not be lower than the total random errors. We see that this empirical upper estimation of total random uncertainty has a median value ($2.8 \times 10^{14}$ molec/cm$^2$) very close to the median total random uncertainty obtained by error propagation theory ($2.6 \times 10^{14}$ molec/cm$^2$), which gives confidence in the overall FTIR error estimation. At all the PROFFIT sites, except the highly polluted one (Mexico city), the total random uncertainty is larger than the Diff30, which could be an indicator that the uncertainty calculated in PROFFIT is slightly too conservative, probably due to this additional channeling error that would be estimated too large. For SFIT4 users, the Diff30 values are usually close, within $0.5 \times 10^{14}$ molec/cm$^2$, to the calculated total random uncertainty, with the exception of Ny-Alesund and Lauder, where the small calculated errors of 1.9 and $1.6 \times 10^{14}$ molec/cm$^2$, respectively, might be a little bit optimistic, and with the exception of Toronto, Wollongong and Paramaribo where 7 to $13 \times 10^{14}$ molec/cm$^2$ differences are observed between the Diff30 values and the total random errors.

After the measurement noise error (and the channeling for PROFFIT users), the largest contributions to the forward model parameters random uncertainty are coming from the temperature, the interfering species, and the off-set baseline. For temperature, the $\mathbf{S}_b$ matrix has been estimated using the differences between an ensemble of NCEP and sonde temperature profiles at Reunion Island, leading to 2 to 4 K in the troposphere and 3 to 6 K in the stratosphere. This matrix is currently used by all SFIT4 users, while for the PROFITT users, these value are chosen smaller (1 K in the troposphere, 2 K up to the middle/upper stratosphere, and 5 K for the highest levels). For each interfering species, the associated $\mathbf{S}_b$ matrix is the covariance matrix obtained with the WACCM v4 climatology. At some stations, the ILS is also a high contribution to the random error budget.

If one uses the FTIR HCHO measurements to validate a model or a satellite with fine-vertical resolution, considering the random and systematic uncertainties (without smoothing) in Table 3 (4th column) is sufficient for making correct comparisons, because the smoothing error due to the low-vertical resolution of the FTIR measurements is vanishing if one takes into account the FTIR averaging kernels and *a priori* profile in the comparisons (Rodgers and Connor, 2003). However, if one wants to have a better knowledge of the real precision of the FTIR data themselves, this smoothing uncertainty can be estimated, for the random part, using the smoothing error covariance $\mathbf{S}_s$ (Rodgers, 2000):

$$\mathbf{S}_s^{\mathrm{rand}} = (\mathbf{I} - \mathbf{A})\mathbf{S}_{\mathrm{var}}(\mathbf{I} - \mathbf{A})^T, \tag{4}$$

where $\mathbf{S}_{\mathrm{var}}$ should represent the natural variability of the target molecule. For HCHO, this $\mathbf{S}_{\mathrm{var}}$ variability matrix is unfortunately not well known due to the poor number of vertically resolved measurements. In Table 3, the smoothing errors have been calculated taking the covariance matrices obtained using the WACCM v4 profiles at each station as an approximation of the $\mathbf{S}_{\mathrm{var}}$ matrices. However, models usually underestimate the variability, and we expect that the smoothing errors provided

here may be underestimated, especially in locations where HCHO is expected to have stronger vertical gradient variability than in the model. As an example, in the study of Vigouroux et al. (2009), the $\mathbf{S}_{\mathrm{var}}$ was taken from aircraft measurements PEM-Tropics-B, and led to a smoothing error estimation of 14% at St-Denis, while the present estimation based on the WACCM model gives about only 2% for this station. However, the $\mathbf{S}_{\mathrm{var}}$ matrix constructed from PEM-Tropics-B showed from 33% to

70% of HCHO variability which seems too much compared to what is observed at Reunion Island from the FTIR measurements (about 20%). This illustrates that, ideally, the $\mathbf{S}_{\mathrm{var}}$ matrix should be re-evaluated at the sites, whenever better model data or correlative measurements become available. The FTIR data sets always including their associated averaging kernel matrices, this can be done *a posteriori* by future users, using Eq. 4.

The smoothing systematic uncertainty, reflecting the bias that would occur on the retrieved profile if the *a priori* $\mathbf{x_a}$ is biased

compare to the real expected profile $< \mathbf{x} >$, is calculated following von Clarmann (2014):

$$\mathbf{S}_{\mathrm{s}}^{\mathrm{syst}} = (\mathbf{I} - \mathbf{A})(\mathbf{x_a} - <\mathbf{x}>)(\mathbf{x_a} - <\mathbf{x}>)^{\mathbf{T}}(\mathbf{I} - \mathbf{A})^{\mathbf{T}}. \tag{5}$$

The $\mathbf{x_a} - <\mathbf{x}>$ is obviously not known (otherwise, $<\mathbf{x}>$ would be chosen as the correct *a priori* in the retrievals). Therefore, we have chosen to use $\mathbf{x_a} - <\mathbf{x}> =$-50%, -20%, -10%, +10%, +8%, +5% for the ground-4km; 4-8km; 8-13km; 13-25km;25-40km; 40-120km layers, respectively. The values have to vary with altitude to induce a different *a priori* profile shape: if 50%

is used at all altitudes, the *a priori* profile is then different from $<\mathbf{x}>$ by a simple scaling factor, and the systematic smoothing error is close to zero. Using the above values, we obtain smoothing systematic errors from 1 to 9% (median value of 3.4%), which is small compared to the other systematic error sources (Table 3). These values assume that the model WACCM profile shapes are not too far from the reality, which should be the case: due to the known short lifetime of HCHO and its production at or near the surface, we expect that the mean profile peaks at the ground. This is, as for the random smoothing part, only

an estimate of the smoothing systematic error. As discussed in von Clarmann (2014), one would prefer even to not give these smoothing errors at all. We prefer to give them to provide to the reader as least an idea of the impact of the smoothing in the precision and accuracy of our FTIR HCHO measurements. But when making model or instrument comparisons, the appropriate use of the averaging kernel and *a priori* profile information, following Rodgers and Connor (2003), allows the user to take implicitly into account the smoothing uncertainty. This means that, for satellite or model comparison, if the methodology

of Rodgers and Connor (2003) is used, there cannot be some different systematic biases at different stations due to different $\mathbf{x_a} - <\mathbf{x}>$.

The dominating systematic uncertainty sources are the spectroscopic parameters: the line intensities and the pressure broadening coefficients of the absorption lines present in our micro-windows. For the HCHO spectroscopic parameters, the linelist in atm16 is following HITRAN 2012 (Rothman et al., 2013), which used the work of Jacquemart et al. (2010), and we use

10% for the three parameters (line intensity, air- and self- broadening coefficients). The larger error source is then the HCHO line intensity parameter, and to a lesser extent the HCHO air-broadening coefficient. In addition, the uncertainties on HCHO columns due to the interfering species spectroscopic parameters are calculated. The dominant ones were found to be due to the pressure broadening coefficients of $CH_4$, HDO, and $N_2O$, with an order of magnitude of about 20% of the error due to the HCHO line intensity.

The other systematic error sources due to forward model parameters are lower or within a few percent (ILS, temperature), except for the PROFFIT channeling source (from 7 to 17%), which also has a systematic component. We see from the Table 3 that the total systematic uncertainty is between 10 and 17% at the SFIT4 stations. For the PROFFIT stations, it lies between 12% and 31%.

## 3  Complete FTIR individual HCHO columns data sets

### 3.1  A network sampling very low to highly polluted levels of HCHO

We show in Fig. 5 the individual HCHO total columns obtained at each station, for a single year only (2016, except for St-Denis: 2011), in order to better see the day-to-day variability. The complete time-series at each station are shown in Supplementary material (Fig. S1). The error bars in Figs. 5 and S1 are the total random uncertainty, i.e. we do not include the systematic errors in order to better visualize the precision of the FTIR measurements compared to the observed day-to-day variability. The FTIR network samples a wide range of concentrations. Indeed we can distinguish first the "clean" sites (shown with the same vertical axis with maximum $15 \times 10^{15}$ molec/cm$^2$) such as the Arctic stations (Eureka, Ny-Alesund, Thule, Kiruna, Sodankyla), the marine stations (Izaña, Mauna Loa, Maïdo, St-Denis, and Lauder; the three former being in addition high altitude stations), and the high-mountain sites (Zugspitze and Altzomoni). These clean sites can have HCHO concentrations at the limit of detection (few $10^{13}$ molec/cm$^2$) with mean values of $10\text{-}25 \times 10^{14}$ molec/cm$^2$ (Table 3), except St-Denis which reaches a mean of $39 \times 10^{14}$ molec/cm$^2$.

Second, we show the intermediate concentration sites (with the same vertical axis with maximum $30 \times 10^{15}$ molec/cm$^2$) such as the tropical coastal site Paramaribo and the mid-latitudes polluted sites in or close to cities and/or vegetation (Peterhof close to St. Petersburg, Bremen, Paris, Boulder). These intermediate sites have mean HCHO total columns of $58\text{-}73 \times 10^{14}$ molec/cm$^2$. The sites with the highest levels of HCHO (vertical axis 50 or $70 \times 10^{15}$ molec/cm$^2$) are Toronto and Mexico City where large anthropogenic emissions are indeed expected (mean of 95 and $221 \times 10^{14}$ molec/cm$^2$, respectively), and places which are also affected by large biogenic emissions such as at Wollongong (mean of $79 \times 10^{14}$ molec/cm$^2$) and the new site of Porto Velho, located at the edge of the Amazonian forest (mean of $188 \times 10^{14}$ molec/cm$^2$).

### 3.2  HCHO diurnal cycle

As explained in the introduction, to reconcile the different results obtained using satellites observing at different time (e.g. SCIAMACHY and GOME-2 measuring in the morning and OMI in the afternoon), it is crucial to have ground-based observations of the HCHO diurnal cycles (Barkley et al., 2013; De Smedt et al., 2015; Stavrakou et al., 2015). The diurnal cycle is also important for model validation, since emissions, chemistry and other processes depend on the time of the day. Our FTIR data set is able now to provide the diurnal cycles at 21 different locations. To separate the effect of the strong seasonal cycle (that will be shown in the next section), we give the diurnal cycle at four different seasons in Fig. 6 for a selection of the sites, while the other ones are provided in Supplemental material (Fig. S2). As seen from Figs. 6 and S2, the diurnal cycles are often

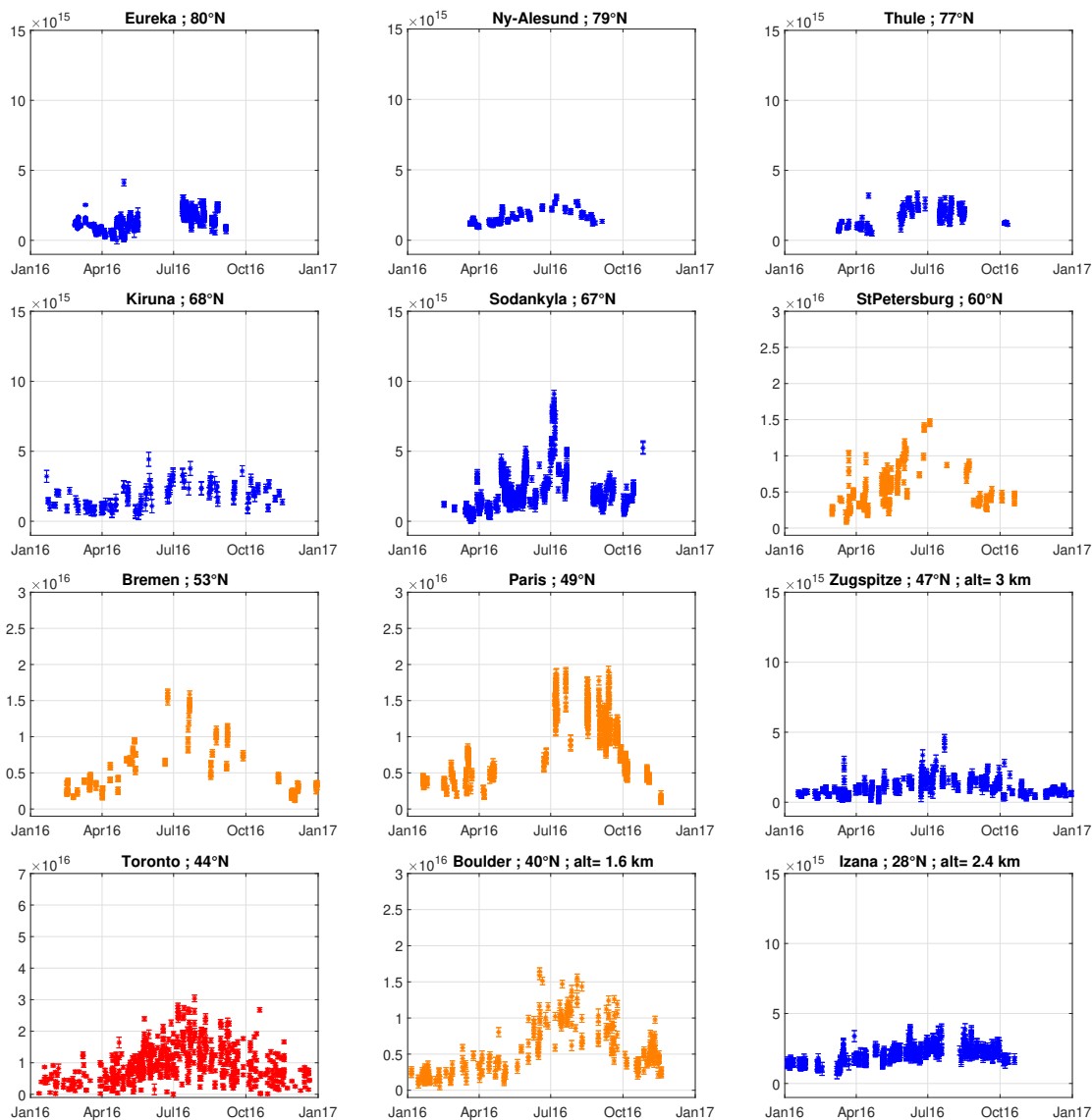

**Figure 5.** Overview of the individual HCHO total columns (molec/cm$^2$) at each station, for a single year (2016, except for St-Denis: 2011). The complete time-series at each station are shown in Supplementary material (Fig. S1) The clean, intermediate, high levels HCHO sites are shown using blue, orange, and red colors, respectively. The error bars are the total random uncertainty. When the altitude of the station is higher than 1.5 km, we explicitly give it.

site and season dependent. While there is no clear diurnal cycle at the Arctic sites and at some of the mid-latitude cities during winter (St. Petersburg, Bremen, Toronto), we usually see an increase from the morning, often more pronounced in June-July-August (and Dec-Jan-Feb in Southern Hemisphere), at most of the stations (in the cities, but also at marine sites). A maximum is often found around midday (St. Petersburg, Mexico City, Izaña, St-Denis, Wollongong), or much later in the afternoon (4-

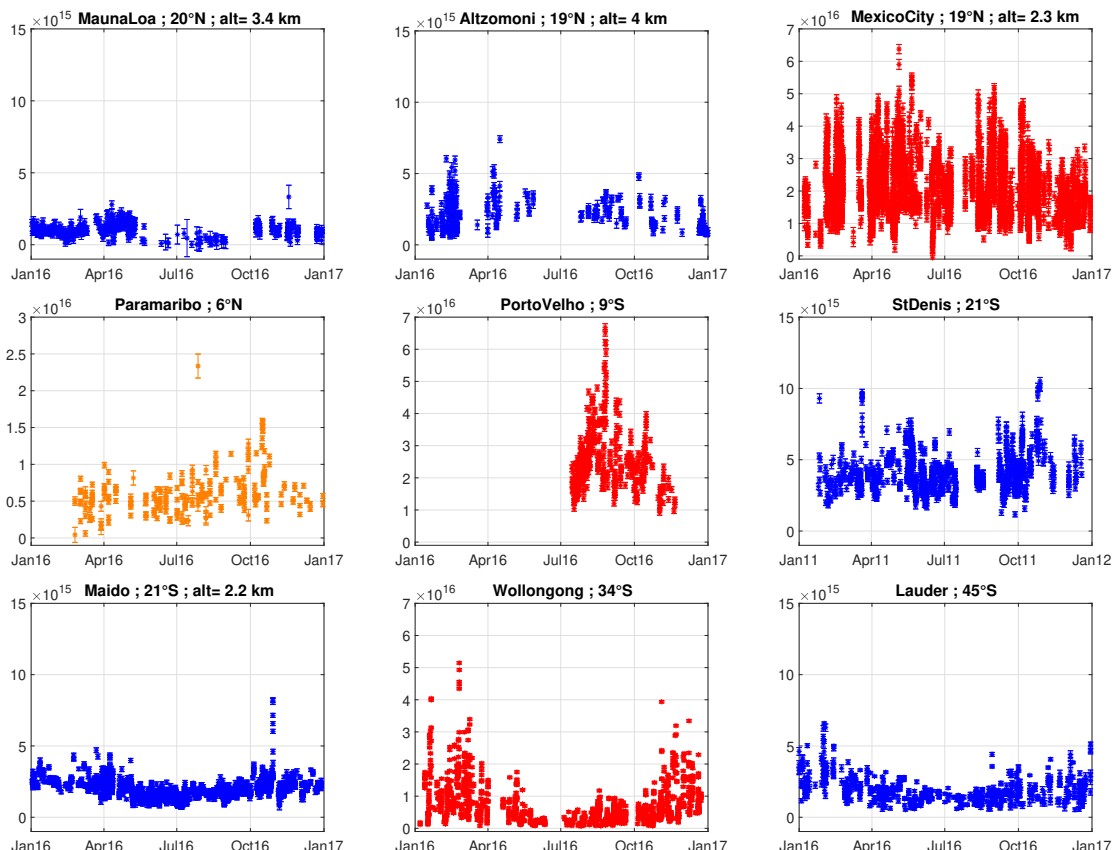

**Figure 5.** *Continued.*

6pm), as in Bremen, Paris, Toronto, Lauder, Altzomoni. Only in a few cases, a minimum is found at midday (St. Petersburg in SON, Zugspitze in MAM-SON, Sodankyla in MAM). The marine sites at high altitudes (free of local pollution) have a small minimum at about 8 am (Izaña, Maïdo). This diversity in the FTIR diurnal cycles is also observed with MAX-DOAS at other sites (De Smedt et al., 2015): very weak diurnal cycle at OHP (Southern France) in Winter and Spring; wide minimum around

5   midday at Beijing and Xianghe in Spring and Autumn, and constant increase in Summer (as observed with FTIR for Bremen, Toronto and Paris). The diurnal cycles observed at the Jungfraujoch station by FTIR measurements (Franco et al., 2015) are showing, for all months of the year, a midday maximum, which is very different from the ones observed at our close station Zugspitze. The IMAGES model shows diurnal cycles in phase agreement with our FTIR measurements at Zugspitze except for the Summer for which the model diurnal cycle is very weak (not shown). However, two very close sites can indeed observe

10   different diurnal cycles (as seen for St-Denis and Maïdo). More investigation is needed to understand the different diurnal cycles at these two close mountain sites.

We see from Fig. 6 that the FTIR measurements at Porto Velho do not show a clear pattern, in particular if one is interested in the 9:30 and 13:30 differences between the overpass of two different satellites (De Smedt et al., 2015). From the one year of

data available at present at this new site, it seems that the diurnal cycle cannot help to reconcile the differences observed over Rondonia between GOME-2 and OMI (De Smedt et al., 2015). In contrast, the diurnal cycles observed over cities confirm that the observation of a positive bias between OMI (13:30) and GOME-2 (9:30) over urban areas can be indeed explained, at least partly, by the diurnal cycle.

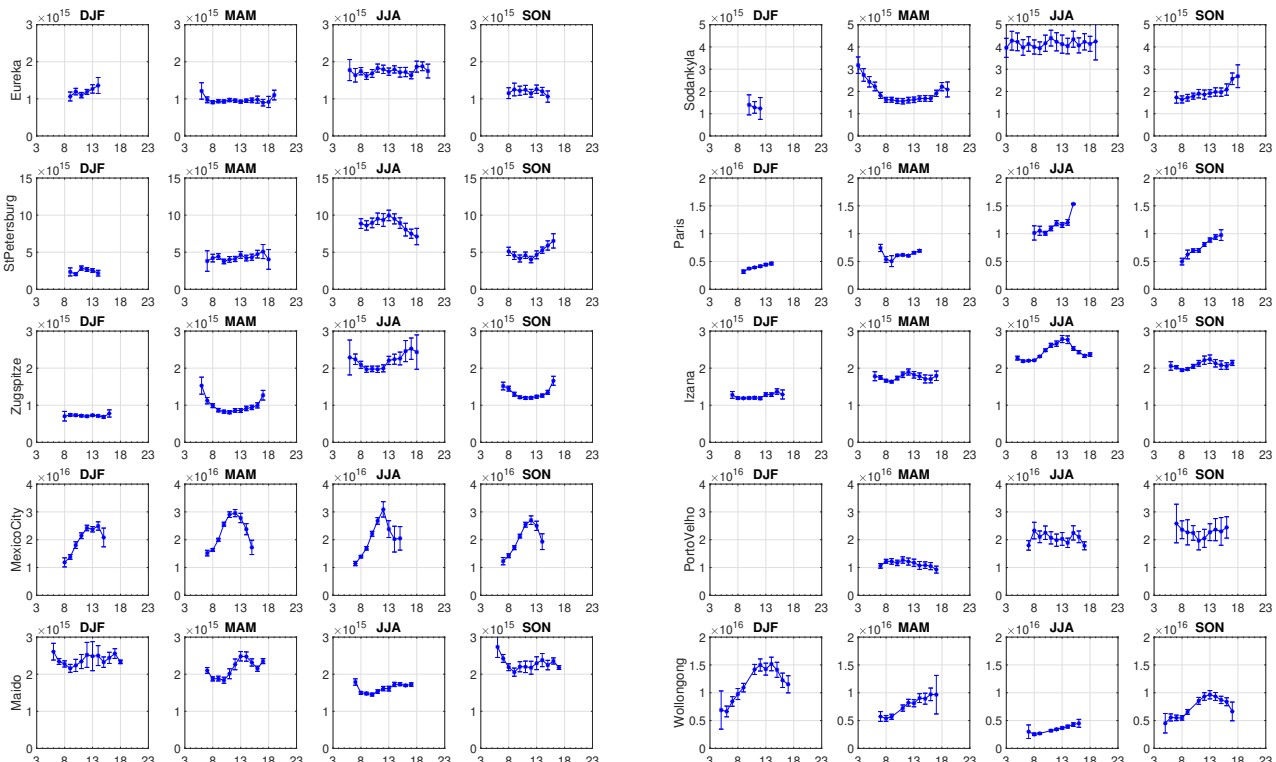

**Figure 6.** Diurnal cycles of HCHO total columns (molec/cm$^2$) at selected stations for the four seasons. The diurnal cycles for the other stations are shown in Supplementary material (Fig. S2). The error bars are the standard errors on the mean: $2 \times \sigma / \sqrt{n}$, with $\sigma$ the standard deviation and $n$ the number of measurements at a given time. If $n < 8$, the hourly value is not shown. The time is the Local Standard Time Meridian (LSTM).

## 3.3 Long-term HCHO trends

The length of the HCHO time-series allows trends to be derived for some stations. We have calculated the trends at each station using the monthly mean time series $Y_m(t)$ with a simple model including a fit of the seasonal cycles:

$$Y_m(t) = A_0 + A_1 \cdot \cos(2\pi t/12) + A_2 \cdot \sin(2\pi t/12)$$
$$+ A_3 \cdot \cos(4\pi t/12) + A_4 \cdot \sin(4\pi t/12)$$
$$+ A_5 \cdot t,$$

with $A_5$ the annual trend.

It turned out that, due to the very high variability of HCHO, the uncertainties on the trends are often too large to obtain significant values. A more sophisticated multi-regression model might be able to reduce the uncertainties, but this is beyond the scope of this paper. However, for a few stations, significant trends are found. They are mainly negative: at St. Petersburg (-3.9 ± 3.3 %/dec), Mexico City (-9.6±5.1 %/dec), Wollongong (-18.8±10.8 %/dec), and close to significance at Zugspitze (-7.7±7.7%/dec). Only the marine sites Izaña and St-Denis show positive significant trends (+17.3 ± 15.2 and +15.8 ± 5.2 %/dec, respectively). Note that at Maïdo, the trend is not significant. A careful combination of the measurements at both Reunion Island sites (St-Denis + Maïdo) could be carried out in the future.

For the longest time-series, we observe in general a very good agreement with previous studies. The negative trends observed over the European stations St. Petersburg and Zugspitze are in agreement with the negative trends observed by OMI (2004-2014) over St. Petersburg and Germany (De Smedt et al., 2015). At the Jungfraujoch station, a negative trend (-6.1±2.6%/dec) was also observed for the 1996-2015 period (Franco et al., 2016). Note that the calculation of the uncertainties on the trends in our study takes into account the auto-correlation in the residuals, following Santer et al. (2000), which increases the uncertainties. For e.g. Zugspitze the uncertainty without correcting for this auto-correlation, as in Franco et al. (2016) or De Smedt et al. (2015), would be 4.9% (instead of 7.7%), showing then a more significant trend, in agreement with these studies. The non significant trends observed at the Northern European station (Kiruna), and the mid-latitude American stations (Toronto, Boulder) are in agreement with De Smedt et al. (2015). In the Southern Hemisphere, the negative trend observed at Wollongong was also found in De Smedt et al. (2015), as well as a positive trend at Madagascar, close to Reunion Island, in agreement with the high positive trend observed at StDenis. At Lauder, OMI also shows non significant trend (De Smedt et al., 2015).

## 4 HCHO FTIR and IMAGES model comparisons

In this study, we do not aim to validate the model input parameters or to attribute different emission sources at the different stations. We use the model to assess the internal consistency of the network using harmonized retrieval settings. This means that we expect that for the same latitude regions and/or type of sites (polluted; clean), the comparisons with the model will give consistent biases.

### 4.1 IMAGES model description

The IMAGESv2 global model calculates the distribution of 170 chemical compounds gases with a time step of 6 hours at $2° \times 2.5°$ resolution, with 40 a hybrid ($\sigma$-pressure) levels in the verticals between the surface and the lower stratosphere (44 hPa level). The model calculates daily averaged concentrations of chemical compounds. The effect of diurnal variations is accounted for through correction factors on the photolysis and kinetic rates obtained from a full diurnal cycle simulation using a time step of 20 minutes. The same model simulation also stores on files the diurnal shapes of formaldehyde columns required for the comparison with FTIR data. Meteorological fields (winds, temperature, humidity, 3-dimensional cloud cover, solid and liquid cloud water content, large-scale and convective precipitation rates, visible downward radiation, convective updraft fluxes,

boundary layer diffusivities, snow depth, sea ice fraction, surface roughness lengths, surface sensible heat flux, friction velocity, etc.) are obtained from ERA-Interim analyses of the European Centre for Medium-range Weather Forecasts (ECMWF).

Anthropogenic emissions of NOx, CO, $SO_2$, and NMVOC are provided by the Hemispheric Transport of Air Pollution dataset version 2 (HTAPv2) (Janssens-Maenhout et al., 2015), with the NMVOC speciation provided by the emission inven-
tory of the Atmospheric Chemistry and Climate Model Intercomparison Project (ACCMIP) (Lamarque et al., 2010). Emissions from open vegetation fires are taken from the last version of the Global Fire Emissions Database, GFED4s, which includes the contribution of small fires (Randerson et al., 2012; Giglio et al., 2013). The GFED data are available at daily frequency at $0.25° \times 0.25°$ from 1997 through the present (http://www.globalfiredata.org). The vertical distribution of these emissions follows Sofiev et al. (2013). Isoprene and monoterpenes emissions are obtained from the MEGAN-MOHYCAN
model (Müller et al., 2008; Stavrakou et al., 2014; Guenther et al., 2012) for all years of the study period at a resolution of $0.5° \times 0.5°$ (http://tropo.aeronomie.be/models/isoprene.htm). Methanol biogenic emissions are obtained from the inverse modeling study of Stavrakou et al. (2011). Besides the dependence on temperature, visible radiation, leaf area and leaf age, the model accounts for the inhibition of isoprene emissions under drought conditions through a dimensionless soil moisture activity factor ($\gamma_{SM}$). However, the parameterization of $\gamma_{SM}$ is very uncertain, as discussed in Bauwens et al. (2016), and we
assume $\gamma_{SM} = 1$ in this study. The average global annual emissions are 419 Tg/yr isoprene, 100 Tg/yr methanol and 103 Tg/yr monoterpenes.

The chemical degradation mechanism of pyrogenic NMVOCs is described in Bauwens et al. (2016). The oxidation mechanism for isoprene is also based on Bauwens et al. (2016), with a few updates. It accounts for the revised kinetics of isoprene peroxy radicals according to the Leuven Isoprene Mechanism version 1 (LIM1) (Peeters et al., 2014) and further modified to
account for laboratory findings (Teng et al., 2017; Bates et al., 2016). The formaldehyde yield in isoprene oxidation by OH is close to 2.4 mol/mol in high NOx (1 ppbv $NO_2$, after 2 months of simulation) and 1.9 mol/mol at low NOx (0.1 ppbv $NO_2$). The chemical mechanism for monoterpenes is simplified, with product yields of formaldehyde, acetone, methylglyoxal and glyoxal based on box model calculations using the $\alpha$- and $\beta$-pinene oxidation mechanism from the Master Chemical Mechanism (MCM) (Saunders et al., 2003). The overall formaldehyde yield is 4.2 HCHO per monoterpene oxidized, coming down to 2.3
after subtracting the contributions of acetone and methylglyoxal oxidation. This yield is further reduced by 45% to account for wet/deposition of intermediates and secondary organic aerosol formation. This fraction of 45% is higher, but of the same order, as the estimated overall impact of deposition on the average HCHO yield from isoprene oxidation (28%), based on IMAGES model calculations. The higher fraction for monoterpenes is intended to account for the impact of the more complex chemistry and larger number of oxygenated intermediates involved in their oxidation, compared to isoprene. The large deposited
fraction is uncertain, but appears justified by the larger number and lower volatility of intermediates involved in formaldehyde formation from monoterpene oxidation.

The calculation of the model columns at the FTIR stations accounts for its location in the horizontal (nearest model pixel), for the FTIR *a priori* profiles and averaging kernels as prescribed in Rodgers and Connor (2003), as well as for the station altitude above sea level. The model column is calculated from the calculated formaldehyde profile, between the altitude of the
station and the model uppermost level (approximately 20 km), and from the *a priori* FTIR profile, above that level. When the

model surface lies higher than the station, the model column is increased by a partial column assuming a constant mixing ratio between the two altitudes, taken equal to the value at the lowermost model level. The monthly averaged formaldehyde columns are calculated by accounting for the temporal sampling of the observations at each site and month. Also, the local time of each observation is taken into account by re-scaling the daily averaged concentration using the formaldehyde diurnal shape factors calculated by the model with a time step of 20 minutes.

## 4.2 HCHO monthly means and seasonal cycle comparisons

We compare the monthly means of FTIR HCHO total columns at each station with the IMAGES columns calculated for the 2003-2016 period. The time-series of both products are shown in Fig. 7. Since the random uncertainty of the FTIR monthly means is divided by the square root of the number of measurements within each month, the dominant contribution to the FTIR error bars in Fig. 7 is the systematic uncertainty (estimated at 12-31%). Except for very few cases (Mexico City and Paramaribo), the model is in overall good agreement in terms of absolute levels (Fig. 7) and seasonal cycle (Fig. 8) with the FTIR measurements.

For each station the correlation, the bias and the standard deviation (std) of the statistical comparisons between the monthly means, mean(IMAGES (smoothed) -FTIR) / mean(FTIR), are summarized in Table 4. The median correlation between FTIR and IMAGES for the 21 stations is very high (0.81), with weaker values at the Mexican stations (0.4/0.5) and at Mauna Loa (0.10). The median standard deviation for all comparisons is 25% (ranging from 11% to 41%). This agreement is good considering the FTIR variability (i.e. the std) of HCHO monthly means (median of 35%). The standard deviation of the comparisons can be explained partly by the lower variability of the model monthly means (31%) compared to FTIR, as seen in Fig. 7. In addition, the variability of the model data within a month is also much smaller (median of about 11%; this STD within a month is shown as magenta error bars in Fig. 7) than the FTIR one (mean of about 28%).

The median of IMAGES and FTIR differences is small (-15%) and within the FTIR systematic uncertainty estimated at 12-31%. However, the biases range from -64% to +51%, which requires an investigation of their possible reasons. The main source of systematic uncertainty is the spectroscopic parameters, which have been harmonized in this work, each station using the same line parameters database, and the same spectral micro-windows. Therefore, it is expected that all FTIR stations should provide consistent HCHO total columns within 5-17% (systematic errors due to other sources than spectroscopic ones). To check this, we divide the FTIR stations according to their concentrations levels and latitudes, and use the model for comparisons.

### 4.2.1 Clean Arctic sites

We distinguish two groups of Arctic sites: Eureka, Ny-Alesund and Thule which are very remote (77-80°N), and two European sites, Kiruna and Sodankyla (67-68°N). As seen in Table 4, the former group shows similar negative biases of the model compared to the data (-20 / -17/ -28%), while the latter group shows positive biases (+32 / +11%). Except at Kiruna, the biases are not constant through the year, the model showing less pronounced seasonal cycles (see also Fig. 8). The model underestimates the summer HCHO levels at the three 77-80°N stations (-26 / -20 / -28%), while the winter levels are in close agreement (+6/ -3%). At the 67-68°N stations, the model is positively biased in winter (+27 / +56%), as well as in summer

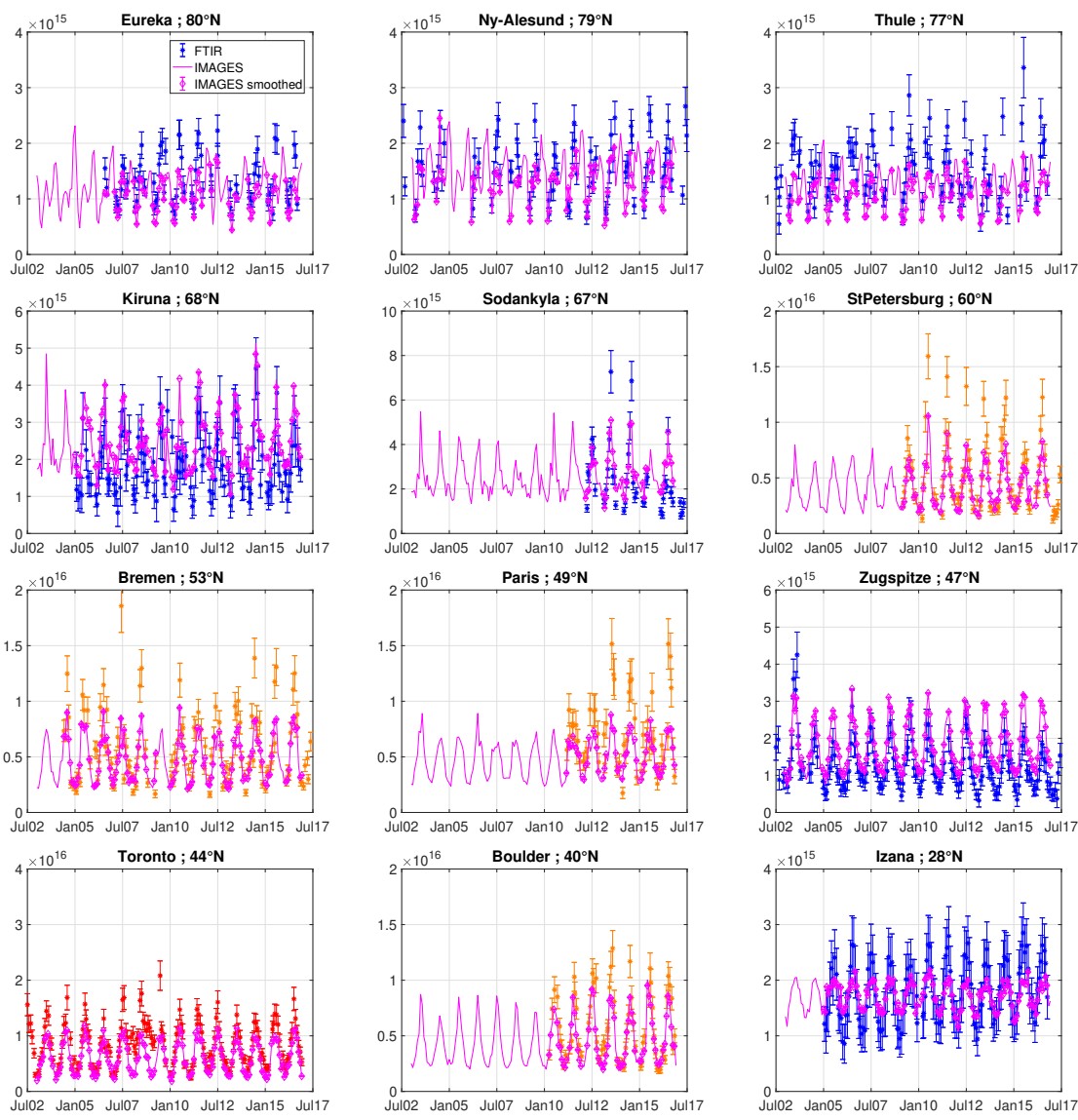

**Figure 7.** Monthly means of HCHO total columns (molec/cm$^2$) at each station for FTIR measurements are shown with stars (clean, intermediate, and high levels HCHO sites are shown using blue, orange, and red colors, respectively) and model data (magenta line for "raw" model data; magenta diamonds for model data smoothed by FTIR AK). The FTIR error bars represent the total uncertainties on monthly means which, due to monthly averaging, are mainly the systematic uncertainties. The model error bars represent the standard deviation of the model for each month.

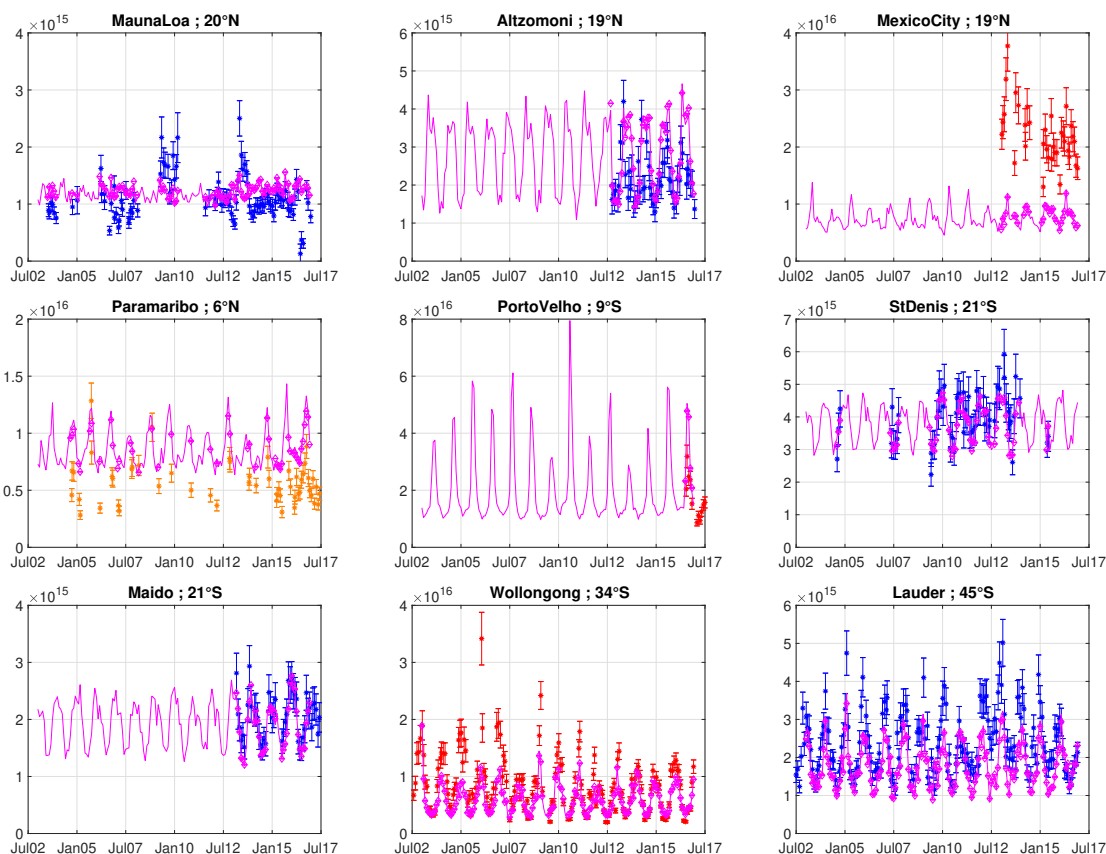

**Figure 7.** *Continued*.

at Kiruna (+22%). Note that the Arctic sites do not have measurement during polar night, so the winter months correspond basically to February (Fig. 8).

### 4.2.2 Mid-latitude cities

Very similar biases (-16 / -15 / -22%) between IMAGES and FTIR are obtained at the three European cities, St. Petersburg (the
5    site is actually at Peterhof, a small coastal city at about 30 km west of St. Petersburg), Bremen, and Paris. As for the Arctic sites, the model underestimates the amplitude of the seasonal cycle (Fig. 8), leading to smaller biases in winter (-14 to -17 %) compared to summer (-19 to -30 %).

     The Northern American sites Toronto and Boulder give similar biases (-26%/-17%), especially in summer (-25%/-17%). Toronto is the only mid-latitude urban site where the model shows a higher underestimation of the HCHO levels in winter
10    (-39%).

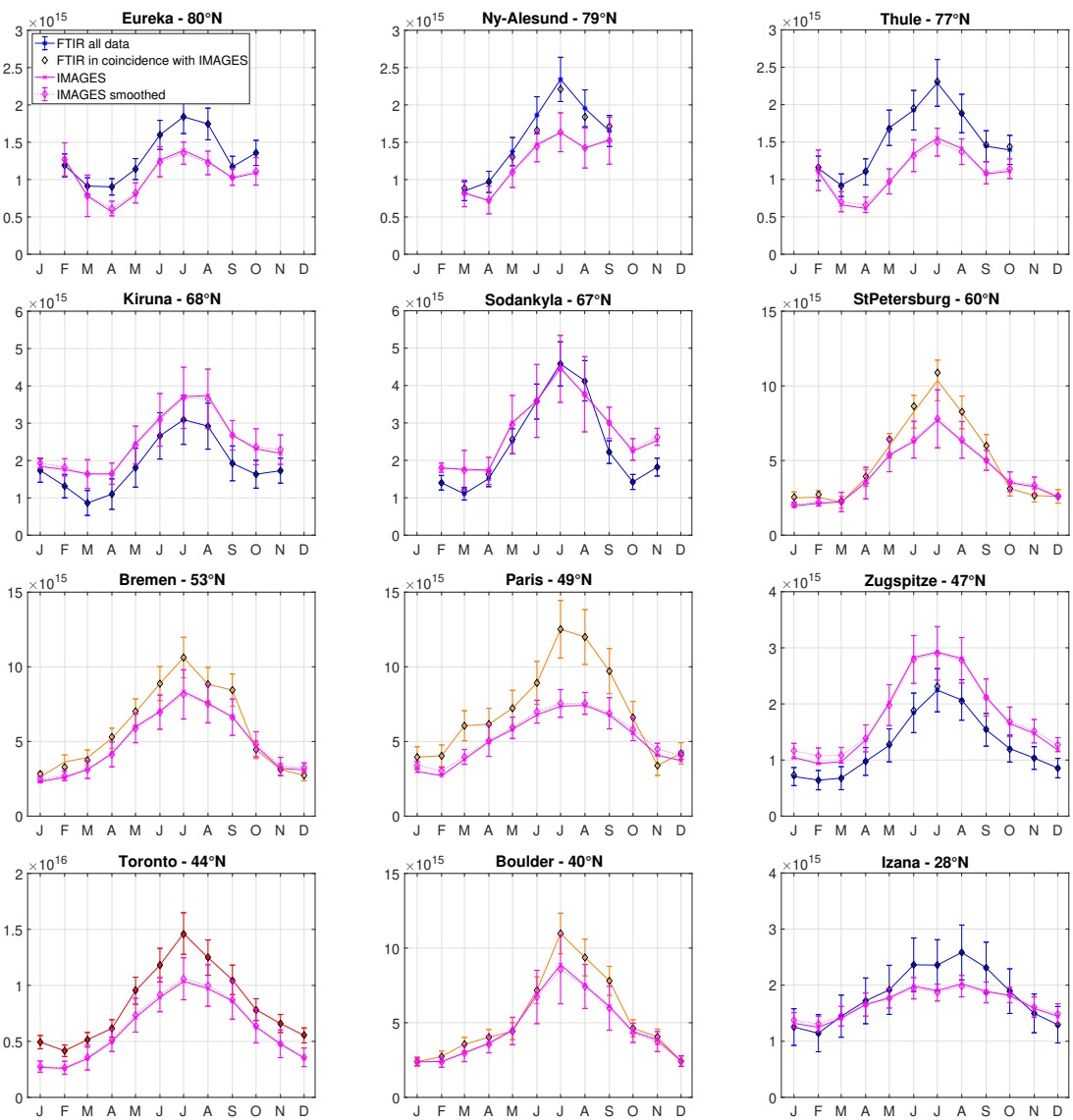

**Figure 8.** Seasonal cycle of HCHO total columns (molec/cm$^2$) at each station for FTIR measurements (clean, intermediate, and high levels HCHO sites are shown using blue, orange, and red stars, respectively, when only data in coincidence with the model are used; black diamonds correspond to the seasonal cycles when all FTIR data are used) and model data (magenta line for "raw" model data; magenta diamonds for model data smoothed by FTIR AK). The FTIR error bars represent mainly the systematic uncertainties. The model error bars represent the standard deviation of the model for each month. Only the model data in coincidence with FTIR measurements are taken into account in these seasonal cycles.

**Table 4.** Correlation (Corr), bias ± standard deviation ($STD_{stat}$) of the statistical comparisons between the monthly means, mean(IMAGES (smoothed) -FTIR) / mean(FTIR). Also given: the mean of the standard deviations in the IMAGES and FTIR monthly means, i.e. the variability within a month ($STD_m$), and the standard deviation of the whole FTIR and IMAGES monthly mean time-series ($STD_{all}$). All numbers, except the correlations, are given in %.

| Station | Corr | bias ± $STD_{stat}$ | bias ± $STD_{stat}$ | bias ± $STD_{stat}$ | $STD_m$ IMAGES/FTIR | $STD_{all}$ IMAGES/FTIR |
|---|---|---|---|---|---|---|
| | All | All | JJA | DJF | Within a month | All |
| Eureka | 0.77 | -20 ± 21 | -26 ± 15 | +6 ± 22 | 10 / 28 | 28 / 32 |
| Ny-Alesund | 0.72 | -17 ± 23 | -20 ± 18 | - | 9 / 25 | 30 / 33 |
| Thule | 0.74 | -28 ± 24 | -28 ± 23 | -3 ± 18 | 9 / 31 | 28 / 35 |
| Kiruna | 0.80 | +32 ± 27 | +22 ± 20 | +27 ± 37 | 10 / 28 | 31 / 44 |
| Sodankyla | 0.85 | +11 ± 33 | -4 ± 27 | +56 ± 35 | 12 / 34 | 37 / 60 |
| St. Petersburg | 0.94 | -16 ± 29 | -25 ± 20 | -14 ± 23 | 12 / 32 | 43 / 60 |
| Bremen | 0.87 | -15 ± 30 | -19 ± 27 | -16 ± 40 | 8 / 20 | 42 / 56 |
| Paris | 0.84 | -22 ± 29 | -30 ± 25 | -17 ± 40 | 6 / 19 | 30 / 45 |
| Zugspitze | 0.87 | +41 ± 26 | +32 ± 24 | +59 ± 24 | 10 / 31 | 37 / 51 |
| Toronto | 0.88 | -26 ± 23 | -25 ± 16 | -39 ± 44 | 15 / 40 | 46 / 47 |
| Boulder | 0.93 | -17 ± 22 | -17 ± 15 | -13 ± 32 | 12 / 24 | 47 / 52 |
| Izaña | 0.81 | -3 ± 20 | -19 ± 9 | +22 ± 15 | 8 / 18 | 14 / 29 |
| Mauna Loa | 0.10 | +13 ± 35 | +14 ± 45 | +24 ± 35 | 9 / 28 | 9 / 34 |
| Mexico City | 0.45 | -64 ± 21 | -59 ± 17 | -66 ± 26 | 17 / 37 | 18 / 23 |
| Altzomoni | 0.43 | +26 ± 41 | +49 ± 22 | -6 ± 22 | 16 / 42 | 35 / 29 |
| Paramaribo | 0.67 | +51 ± 25 | +59 ± 15 | +85 ± 17 | 12 / 35 | 17 / 33 |
| | | | DJF | JJA | | |
| Porto Velho | 0.87 | +41 ± 35 | - | +35 ± 35 | 24 / 27 | 39 / 26 |
| St-Denis | 0.71 | -7 ± 13 | -3 ± 12 | -9 ± 15 | 9 / 27 | 16 / 18 |
| Maïdo | 0.87 | -7 ± 11 | +3 ± 7 | -14 ± 7 | 13 / 20 | 23 / 20 |
| Wollongong | 0.83 | -26 ± 37 | -29 ± 34 | -3 ± 35 | 18 / 50 | 43 / 59 |
| Lauder | 0.77 | -25 ± 22 | -24 ± 17 | -26 ± 26 | 11 / 31 | 31 / 35 |
| | | | "Summer" | "Winter" | | |
| Median | 0.81 | -15 ±25 | -19 ±19 | -5 ±26 | 11 / 28 | 31 / 35 |

### 4.2.3 High-mountain sites

The mountain sites are more difficult to model especially when they are close to cities. They are often very clean sites, but the model cannot reproduce this at the current resolution ($2° \times 2.5°$) when they are surrounded by emission sources in the same

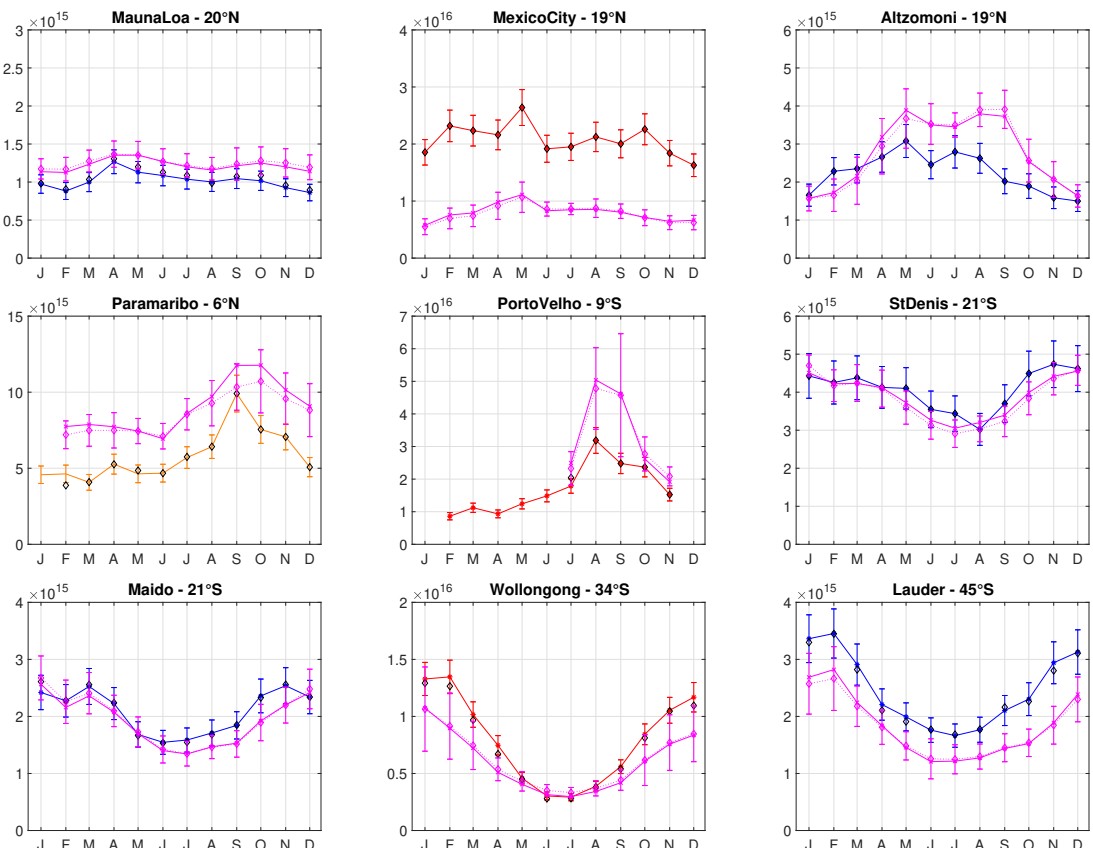

**Figure 8.** *Continued.*

pixel. This seems to be the case at Altzomoni, which lies in the same model pixel as Mexico City, leading to an overestimation of 26%, much larger in summer (+49%), and at the European station Zugspitze where the model overestimates the HCHO levels by +41%. Note that in the study of Franco et al. (2015), a negative bias (-13%) was observed between FTIR at Jungfraujoch (47°N, 8°E) and IMAGES, but the retrieval settings used were different than in the present study. Only a change in the spectroscopic

5 database, from HITRAN 2008 to HITRAN 2012, led to lower HCHO columns by 49% at Jungfraujoch (Franco et al., 2015). It is therefore not possible at present to compare the biases obtained at these two close stations.

At the mountain site of Izaña, located at a clean marine area, the model and FTIR are in overall good agreement (-3%), with a negative bias in summer (-19%) and a positive one in winter (+22%), as a result of the weak seasonal amplitude in the model. A moderate positive model bias is calculated at Mauna Loa (+13%), more pronounced in winter (+24%), and a good agreement

10 is seen between the model and FTIR mean seasonal cycle (Fig. 8). The observed variability (34%) is however important at this site, and similar to e.g. the clean Arctic sites (Fig. 7), with values ranging from 0.5 to $2.5 \times 10^{15}$ molec/cm$^2$. This is not reproduced by the model values lying within $1$-$1.5 \times 10^{15}$ molec/cm$^2$. The reasons of the pronounced observed variability are unclear at present."

#### 4.2.4 Central and South American sites

The model falls short in reproducing the enhanced HCHO levels observed at Mexico City ($ca\ 2 \times 10^{16}$ molec/cm$^2$), mainly due to the coarse model resolution($2° \times 2.5°$), as suggested by the strong negative bias (-64%), which is almost constant across the year.

Comparison at two sites in South America, the coastal site of Paramaribo and the Porto Velho site at the edge of the Amazonian forest, indicates a consistent model overestimation (+51 / +43%). At Porto Velho, this overestimation is more significant during the dry season (August-September, Fig. 8), which corresponds to the maximum of fire intensity in Amazonia. An overestimation of biogenic (isoprene) and biomass burning emissions in Amazonia was already found in IMAGES in the study of Bauwens et al. (2016).

#### 4.2.5 Southern Hemisphere 21-45°S sites

The two marine sites at Reunion Island (St-Denis at sea level, and Maïdo at 2.2 km altitude) show a small model bias (-7%) and standard deviation, especially at Maïdo (11%). At these sites, HCHO shows the lowest variability in the monthly means (18-20%), and the model reproduces quite well the seasonal cycle. As shown in Fig. 8, the largest seasonal bias is not found in austral summer (DJF) as seen in the Northern Hemisphere sites, but during September-November months, which correspond to the maximum of the biomass burning period in Southern Africa and Madagascar, close to Reunion Island. The biomass burning source at this location might be underestimated, while it was overestimated in South America.

The Wollongong site shows the same behavior as most of the Northern Hemisphere sites: an overall underestimation of the model (-26%), larger in austral summer (-29%). A first look on the Lauder comparison gives a similar annual bias (-25%), which remains constant over the year, as seen in Table 4 and Fig. 8. However, Fig. 7 shows that during the austral winters (JJA) 2012 to 2015, the FTIR time-series presents unusually high columns. By limiting the comparison to the first years of the period, a better agreement with the model in winter is obtained at Lauder as often observed at other sites.

Since the time-series at St-Denis, Wollongong and Lauder have been published in the past using different retrieval strategies (Vigouroux et al., 2009; Jones et al., 2009; Zeng et al., 2015), we report here the bias observed at these stations between the previous and present data sets. The bias at St-Denis between the previous data set using the strategy in Vigouroux et al. (2009), in which the *a priori* profile and the spectroscopy were different (mostly for interfering species, the HCHO spectroscopic intensity parameters being from the same work of Perrin et al. (2010)), and the mws were smaller than the present work, is only of 1.4% (the new HCHO columns being smaller). Therefore, the comparisons with MAX-DOAS shown in Vigouroux et al. (2009) would still provide a good agreement between the two techniques. Concerning Lauder and Wollongong, where the previous retrieval strategy was from Jones et al. (2009), the present HCHO columns are 49% smaller than the previous data sets. Therefore, the new data set is in much closer agreement with the simulation of four different models that were all of them found 50% lower than the old Lauder and Wollongong data sets (Zeng et al., 2015). From performed sensitivity tests, this high bias between the two strategies is very likely mostly due to the 2869.65-2870.0 cm$^{-1}$ window used in Jones et al. (2009).

## 5 Conclusions

Only five NDACC FTIR sites delivered HCHO time-series until now (Paton-Walsh et al., 2005; Jones et al., 2009; Vigouroux et al., 2009; Viatte et al., 2014; Franco et al., 2015), using different retrieval settings. The small number of stations and the bias differences associated with the different retrieval strategies made it difficult to use the FTIR network as a coherent tool for satellite or model validation. In this study, we have designed a harmonized HCHO retrieval strategy to derive total columns at 21 stations, at locations characterized by very different concentrations, from very clean Arctic sites where HCHO is at the limit of detection (a few $10^{13}$ molec/cm$^2$) to highly polluted sites such Mexico City or Porto Velho, near the Amazonian forest, where columns up to $7\times10^{16}$ molec/cm$^2$ have been observed. This network includes well-established NDACC stations, as well as several new sites (Sodankyla, Boulder, Paris, Porto Velho) that aim to be affiliated with NDACC. The FTIR network is also growing, with new sites such as Hefei in China, which will again expand its spatial coverage.

We have presented the retrieval settings that have been optimized for this challenging species, and the FTIR HCHO products have been characterized by their averaging kernels, and their uncertainty budget. The systematic uncertainty of an individual HCHO total column measurement lies between 12 and 27%, with still some differences between the SFIT4 code users (12-15%) and the PROFFIT users (12-27%), which needs to be investigated in the future within the NDACC InfraRed Working Group. The random uncertainty lies between 1 and $11\times10^{14}$ molec/cm$^2$, with a median value of $2.9\times10^{14}$ molec/cm$^2$, the high maximum value being due to the lower quality of the Bruker Vertex compared to the high resolution ones (Bruker 120/5M or 120/5HR).

In addition to the well-defined seasonal cycles, the diurnal cycles were presented at each site. These observations are crucial to interpret the differences observed between satellites measuring at different local times. For example, the diurnal cycle at Porto Velho which shows insignificant variations suggests that the negative bias observed over Rondônia between OMI (13:30) and GOME-2 (9:30) (De Smedt et al., 2015) is unlikely due to the diurnal cycle. In contrast, the FTIR diurnal cycles in the cities confirm that the positive bias between OMI and GOME-2 over urban areas is likely due, at least partly, to the diurnal cycle.

The monthly mean time-series as well as the seasonal cycles have been compared to the IMAGES model. We did not aim at evaluating the model, but at showing that the FTIR network provides coherent absolute values and seasonal cycles. We observed an overall good agreement with IMAGES, the model usually (but not always) underestimating the HCHO total columns (median bias $\pm$ standard deviation of -15% $\pm$ 25%), with a more pronounced bias during summer (-19% $\pm$ 19%). The similar biases obtained at stations under similar conditions (clean Arctic sites, urban sites, marine sites) strengthen our confidence in the harmonization of the HCHO products within the network. When the model showed different behavior for some of the stations, we could explain it by either the too large size of the model pixel ($2.0° \times 2.5°$), especially for high-altitude sites, as in Zugspitze, Altzomoni, Mexico City; or an overestimation of the biogenic and biomass burning sources in South America (Paramaribo, Porto Velho), which was already pointed out in Bauwens et al. (2016). However for a few sites, the behavior of the model remained unexplained (positive biases at Kiruna and Sodankyla, the too low model variability at Mauna Loa).

These HCHO time-series, harmonized and well-characterized, provide an important data set for past and present satellite, and model validation. They are continuously extended by new measurements and will be used in the coming years for the validation of new satellites, such as Sentinel 5P, and Sentinel 4.

*Acknowledgements.* This study has been supported by the ESA PRODEX project TROVA (2016-2018) funded by the Belgian Science Policy Office (Belspo). NCAR is supported by the National Science Foundation. The NCAR FTS observation programs at Thule and Mauna Loa are supported under contract by the National Aeronautics and Space Administration (NASA). The Thule work is also supported by the NSF Office of Polar Programs (OPP). We wish to thank the Danish Meteorological Institute for support at the Thule site and NOAA for support of the Mauna Loa site. Eureka measurements were made at the Polar Environment Atmospheric Research Laboratory (PEARL) under the CANDAC and PAHA projects led by James R. Drummond, and in part by the Canadian Arctic ACE/OSIRIS Validation Campaigns, led by Kaley A. Walker. Funding was provided by AIF/NSRIT, CFI, CFCAS, CSA, ECCC, GOC-IPY, NSERC, NSTP, OIT, PCSP, and ORF. Logistical and operational support was provided by PEARL Site Manager Pierre Fogal, the CANDAC operators, and the ECCC Weather Station. Toronto measurements were made at the University of Toronto Atmospheric Observatory, supported by CFCAS, ABB Bomem, CFI, CSA, ECCC, NSERC, ORDCF, PREA, and the University of Toronto. The measurements at Reunion Island have been also supported by the Université de La Réunion and CNRS (LACy-UMR8105 and UMS3365), and at Porto Velho by the BRAIN-pioneer project IKARE, funded by Belspo. The measurements at Paramaribo have been supported by the BMBF (German Ministry of Education and Research) in the project 5 O3CHEM (01LG1214A). We thank the Meteorological Service Suriname for support. The measurements and data analysis at Bremen are supported by the Senate of Bremen. The measurements at the St. Petersburg site (SPbU) have been supported by the Russian Science Foundation (project #14-17-00096). Observational facilities have been provided by the Centre for Geo-Environmental Research and Modelling "GEOMODEL" of SPbU. Analysis of FTIR data acquired at SPbU has been performed with the financial support of the Russian Foundation for Basic Research (project #18-05-00011). The measurements at Lauder are core-funded by NIWA, through New Zealand's Ministry of Business, Innovation and Employment. We are grateful to Sorbonne Université and Région Île-de-France for their financial contributions as well as to Institut Pierre-Simon Laplace for support and facilities. The Altzomoni and Mexico City measurements have been funded by DGAPA, PAPIIT (No. IN112216 and No. IN111418) as well as CONACYT (No. 275239 and No. 239618). The German partners acknowledge BMWi for support in HCHO data analysis. The authors would like to thank essential people for the FTIR measurements (C. Hermans, N. Kumps, F. Scolas, M. Zhou, BIRA-IASB; C. Morais, IFRO; U. Raffalski, IRF; E. Sepulveda, AEMET; S. Mitro: Meteorological Service of Suriname; P. Jeseck, LERMA-IPSL; A. Bezanilla, O. Lopez, M. Angel Robles, A. Rodriguez Manjarrez, D. Flores Roman: CCA-UNAM). We thank the station personnel at the AWIPEV research base in Ny Alesund, Spitsbergen, for performing the measurements. We also thank the AWI Bremerhaven for logistical support.

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
