# Peer review of "NDACC harmonized formaldehyde time-series from 21 FTIR stations covering a wide range of column abundances"

_Atmospheric Measurement Techniques, 2018_

## Referee Comment (RC1) · Anonymous Referee #1 · 5 Mar 2018

This paper describes the production of a harmonised data set for HCHO column abundances from 21 FTIR stations located across the globe. First of all, I must commend the authors for pulling this off. I cannot imagine it was an easy task. Bringing together the HCHO measurements from these different stations/groups is an important achievement, and it will be a valuable resource for modelling studies and for satellite validation purposes. It is a great step forward. I urge the authors to create an online repository where the data can be download easily by others.

Overall I recommend the paper for publication, the science and methods are sound, the results important, and it is well-written. I only have minor comments that need

clarifying.

Minor comments

Please ensure all figures are large in the final version of the manuscript – they are very small and difficult to resolve; it is frustrating. Maybe they could be enlarged by breaking them into separate figures

I noticed that in Table 1 the observing period is very long for some of these stations (e.g., Ny-Alesund). Could the authors possibly comment on the instrument stability and performance over such long periods, and if it affects the HCHO retrievals at all? HCHO is difficult to retrieve is not?

Page 7, line 25. The a priori HCHO profile. The approach used here seems sensible, but how sensitive are the retrieval total columns to the a priori – especially as the DOFs is low.

Figure 3. The use of atm16 is clearly necessary; HITRAN 2012 needs some corrections….

Table 3: The 'DIFF30' is a useful metric, it is given in %, but relative to what? Please be explicit. I'm actually surprised its values are so low (<25%) which is encouraging. Can you also indicate which sites are PROFFIT.

Page 13, lines 14-16. Some units are missing?

Page 13, last line. Variability faster than 30 mins. Is there any evidence for this in the literature (e.g., from models, campaigns).

Page 14, line 34: Typo – capital needed at ".this matrix…"

Page 15, line 7, Typo. "1E13" should be x1013

Figure 5: I can understand why this has been plotted but I think it would also probably be good to show the individual measurements for a single (common) year, rather than

[Figure]

over the entire time record at each location. That way you can look at the day to day variability more closely – maybe put such a figure in the supplementary material.

Figure 6. The variability in the HCHO observations poses some interesting questions. There is a lot of science in here.

Page 19. The 45% yield reduction – this maybe indeed correct – but can you provide a little more explanation/justification.

Page 23: High mountain sites – at such locations it might be wise to quantify the difference between the station elevation and the model elevation for the 2x2.5 degree grid cells (maybe add information to Table 4). Is there any correlation between this difference and the bias?

---

## Referee Comment (RC2) · Anonymous Referee #3 · 26 Mar 2018

This study presents the retrieval settings of formaldehyde from ground-based FTIR solar spectra, which has been harmonized to allow for consistent retrievals at various stations, under various conditions (remote area, polluted sites, high-altitude sites...). An error budget is presented for each station. The formaldehyde times-series are then presented along with a preliminary investigation of trend and diurnal cycles. Finally, the consistency of the FTIR products is evaluated via comparison with formaldehyde columns simulated by a chemistry-transport model.

Developing harmonized formaldehyde retrieval settings through the NDACC (and future affiliated stations) is quite challenging because of the weak absorptions of HCHO

in the infrared and the many interferences. This work is therefore valuable in the framework of validation efforts of model simulations and of current and future satellite instruments. The topic developed here fits perfectly the scope of AMT. The paper is globally well written and the structure is clear. Nonetheless, some results/figures are not adequately presented, which somehow impedes a proper evaluation of the results (see major comments here below). Therefore, I recommend publication of this study after addressing the comments listed hereafter.

Major comments

Additional effort is needed to present the results more synthetically and to make some figures easier to read. Fig. 5-8 are particularly difficult to read because of the numerous small panels. It is really unfortunate because these figures present the main results of the study. I assume that the large number of subplots makes them difficult to display, but I really think that such figures deserve a reshape. In particular:

- The various x-axes in Fig. 5 and 7 don't help the reader. Some seasonal cycles appear completely squeezed because these panels encompass >15 years, while others represent 2-3 years only on a panel of the same size. Please display the time series on an x-axis common to all the stations.

- For Fig. 5 and 7, I also suggest to gather some time-series within the same panels, e.g., following the subdivision in the text (i.e. clean, intermediate, polluted sites), using different colour lines. The interest to gather time series within the same panels would also be to help the reader to appraise the large panel of HCHO columns covered by the FTIR sites.

- For Fig. 6, perhaps it is not needed to display all the diurnal cycles in the manuscript. A solution would be to keep here only a few representative of those to support the discussion. The others can be moved to supplementary material.

- In Fig. 7-8, it is very hard to distinguish the raw model data from the smoothed ones.

[Figure]

I also find the discussion on the basis of Table 4 quite "raw". The authors made huge efforts to harmonize the retrievals and to produce a consistent pattern of HCHO measurements worldwide. There could be, along with Table 4, a map including in colour background the mean HCHO from the model over 2003-2016, and the mean FTIR HCHO in dots filled following to the same colour bar, at the location of each station. In one glance, the reader would have a good overview of the pattern of FTIR measurements as well as of the overall consistency with the model.

A single HCHO a priori profile is used at each station for the retrievals. This assumes that not only the shape, but also the HCHO concentration simulated by the model, are quite reliable. What is the impact of another HCHO a priori profile on the retrieved columns? e.g., an a priori from another model that would be significantly different, or again an a priori that would be derived from other measurements (like ACE-FTS)? For a very weak absorber like HCHO, with little retrieved vertical information, I expect the impact to be, if not substantial, at least not negligible. For example, it is clear from the very similar shapes of the a priori and retrieved profiles in Fig. 4, that the retrievals are dependent on the a priori. It is important to discuss this point and to add this component to the error budget.

The error budget is established for each station on the basis of a single measurement. Why one measurement only per station? I find this very reductive, especially that it is not even said whether this single measurement is representative for the whole data set (DOFS, residuals, total column...). Hence, one can easily casts doubts about such error budget. This should be made ideally with a representative subset of FTIR measurements, covering different seasons, different zenith angle, etc.

Page 14, Lines 9-16: Since you know that models usually underestimate the natural variability of HCHO, and since you know the impact of such underestimation on the smoothing error estimation, why wouldn't you increase the variability of the model to be more representative for the real variability? Knowing the difficulty for the models to simulate highly-variable reactive gases like HCHO, a model variability multiplied by
e.g., 2, would still be conservative.

Page 15, Lines 15-27: The diurnal cycle is sometimes very weak. Furthermore, the midday observations (low zenith angle) probing less atmosphere, we can expect larger uncertainties associated with such measurements. Hence, owing to these larger midday uncertainties, are the diurnal cycles still significant? Or couldn't some of these diurnal cycles (e.g., the midday minimum found at some sites) be just the effect of larger uncertainties and less sensitivity associated with the low zenith angle?

Section 3.3: The investigation of the trends are very preliminary, and there is no discussion of the results. If, as quoted in this section, a more comprehensive investigation is beyond the scope, I don't see the interest of this section as it is currently. Or I recommend to add a bit of discussion, e.g., how do significant trends compare to other trends in the literature from, e.g.:

- De Smedt et al. (2010) Trend detection in satellite observations of formaldehyde tropospheric columns, Geophys. Res. Lett., 37, L18808, doi:10.1029/2010GL044245.

- De Smedt et al. (2015) Diurnal, seasonal and long-term variations of global formaldehyde columns inferred from combined OMI and GOME-2 observations, Atmos. Chem. Phys., 15, 12519–12545, doi:10.5194/acp-15-12519-2015.

- Franco et al. (2016) Diurnal cycle and multi-decadal trend of formaldehyde in the remote atmosphere near 46° N, Atmos. Chem. Phys., 16, 4171-4189, https://doi.org/10.5194/acp-16-4171-2016.

- Jones et al. (2009) Long-term tropospheric formaldehyde concentrations deduced from ground-based fourier transform solar infrared measurements, Atmos. Chem. Phys., 9, 7131–7142, doi:10.5194/acp-9-7131-2009.

There have already been comparisons between previous HCHO columns from the FTIR and from UV-Vis instruments (satellites and MAX-DOAS), showing in overall a good agreement. However, it is obvious from this study that the HCHO retrievals from

the FTIR are very sensitive to the retrieval choices (spectroscopic database, micro-windows, a priori...). Biases up to 50 % are even mentioned between different retrieval approaches. This means that the harmonized retrievals presented here can potentially improve or deteriorate significantly the comparisons with the UV-Vis instruments. I think this point needs to be discussed, or at least mentioned in the conclusion.

Minor comments

Page 1, Line 3: "accurate and precise". In light of the error budget and the large biases depending on the retrieval choices ("as large as 50%", Line 5), this statement should be dampened.

Page 1, Line 8: stations. Most of them

Page 1, Line 11: Change ";" to ","

Page 2, Line 1: Unclear. Is it for the systematic or the random uncertainties?

Page 2, Line 11: NOx is not defined yet

Page 2, Line 16: of only a few hours

Page 2, Line 17: and to test

Page 2, Line 32: at a few locations

Page 3, Line 3: Change "geographical" by "spatial"

Page 3, Line 3: A lot of efforts are

Page 3, Line 7: stations that will also be part

Page 3, Lines 13-16: Isn't it because HCHO is so challenging to retrieve that it is not (yet?) a standard product from the NDACC FTIR?

Page 3, Line 27: monthly mean time-series

Page 4: Fig. 1 would deserve to be a bit larger due to e.g., the concentration of stations

in Europe

Page 4, Line 14: (1995) and/or Hase et

Page 4, Line 15: pressure- and temperature-dependent

Page 5, Lines 5-6: It is said elsewhere in the manuscript that the use of different retrieval parameters can substantially affect the retrieved columns. In particular, the use of either HITRAN 2004, 2008 or 2012 for the HCHO spectroscopic parameters leads to very large differences in the retrieved columns. I would have expected the authors to better motivate their choice of HITRAN 2012, especially that eventually some lines of interfering species needed to be empirically adjusted in this spectroscopic database.

Page 6, Line 3: that is distributed

Page 7, Line 8: Rodgers (2000)), is

Page 7, Lines 6-9: Is the little gain in information the only reason why you keep these two windows?

Page 7, Lines 10-11: Is this individual spectrum representative for the whole time-series? How do its DOFS and its fitting residuals compare to the other observations?

Page 7, Line 26: WACCM v4 (Garcia et al., 2007).

Page 7, Line 29: not to use

Page 7, Line 33: Sussmann et al. (2011), and

Page 8, Fig. 2 caption: total column of

(the same in Fig. 3 caption)

And further, same line: Change "The figures in the lower panel are" to "The lower panels are"

Page 10, Line 31: DOFS, in Table 3

[Figure]

Same line: provide more than

Page 10, Line 33: (upper panels)

Page 11, Fig. 4 caption: Upper panels: averaging

And further: Lower panels:

Page 11, Line 5: associated with

Page 11, Line 6: (lower panels)

Page 11, Fig. 4: Do you have an explanation for the contribution from the high-altitude averaging kernels (in green)? It looks a bit odd to have a contribution from such layers to the HCHO retrievals. Still Fig. 4: From the shape and the high values of the total column averaging kernel, don't you "overfit" the HCHO retrievals?

Page 12, Table 3, as well as in the manuscript: Providing the random uncertainties in total column only is really misleading, especially in the discussions. Each time, it forces the reader to look at Table 3 and to calculate the percentage before knowing whether it is significant for the station that is considered. I recommend to provide all the uncertainties (also) in percentage of the total column (you already do it for the systematic uncertainties).

Page 13, Lines 25-29: Why such exceptions for these SFIT4 stations, and not for others? Isn't the error budget supposed to be fully harmonized among all the SFIT4 stations?

Page 13, Line 29: might have

Page 13, Line 33: stratosphere. This matrix

Page 13, Line 34: while for the PROFFIT users, these values

Page 14, Lines 3-4: rephrase as "considering the random uncertainty in Table 3 (4th column) is sufficient" to avoid misleading

Page 14, Line 24: HCHO line intensity.

Page 14, Line 26: the PROFFIT channelling source (from 7 to 17 %), which also has a systematic component. We see from Table 3

Page 15, Line 5: few 1 x 1013 molec/cm$^2$

Page 15, Lines 15-16: Bad sentence. "To reconcile the different . . . (afternoon), it is crucial to have ground-based . . .

Page 15, Line 21: mid-latitude cities

Page 18, Lines 1 and 5: The use of "time step" is here misleading with the computational time step of the model. I suggest to use "with outputs every 6 hours/ 20 minutes" instead.

Page 18, Line 11: delete one "the"

Page 18, Line 22: in Bauwens et al. (2016)

Page 19, Line 1: justified by the

Page 19, Lines 3-6: How do you deal with the model surface that is below the altitude of the station (which should be the case for most of the mountain sites), especially where there is a substantial altitude difference?

Page 19, Lines 7-9: Do you mean that you re-scale the model outputs at the time of the FTIR measurements? Or do you use the nearest model output to each FTIR data?

Page 19, Line 24: "within a month" is redundant

Page 19, Line 25: "The median of IMAGES and FTIR differences" or "The median of IMAGES and FTIR biases"

Page 19, Line 30: change ";" to ","

Page 21, Fig. 8 caption: "in coincidence with". Do you mean the same day, within 20

minutes of each FTIR data?

Page 23, Line 13: Is Boulder ($\sim$1600 m asl) really a urban site?

Page 23, Lines 29-30: At such a remote site, the dominant source of HCHO should be CH4 oxidation. I don't think that other sources from continental areas can be significantly at play here.

Page 24, Lines 5-9: Could it be also due to the FTIR technique, which measures in clear-sky conditions only? The FTIR would sample only air masses free of huge emissions and hence would underestimate the gas abundance in this region.

Page 25, Lines 14-15: We do not aimed at evaluating the model, but at showing that the FTIR

I have here an open question, which I think is relevant for a data set that is designed to be used for intense model and satellite validation efforts. Will this data set be made publicly available? And if yes, will there be an effort to fully harmonize the archives, the file format, and the way the data are saved? There are currently inconsistencies between FTIR data sets from different stations (especially for the AKs). Such inconsistencies sometimes refrain external users to use NDACC FTIR data, while such data sets deserve to be easily accessible and as user-friendly as possible for non-community users.

---

## Author Comment (AC1) · 25 May 2018

This paper describes the production of a harmonised data set for HCHO column abundances from 21 FTIR stations located across the globe. First of all, I must commend the authors for pulling this off. I cannot imagine it was an easy task. Bringing together the HCHO measurements from these different stations/groups is an important achievement, and it will be a valuable resource for modelling studies and for satellite validation purposes. It is a great step forward. I urge the authors to create an online repository where the data can be download easily by others. Overall I recommend the paper for publication, the science and meth-

[Figure]

**ods are sound, the results important, and it is well-written. I only have minor comments that need clarifying.**

We warmly thank the referee for his/her kind supporting words. Concerning an online repository where the data can be downloaded, this is currently under discussion within the InfraRed Working Group (IRWG) community, and a final decision will be taken at the next IRWG meeting in June. The data will be very likely downloaded in the public NDACC repository. In the meantime, the whole data set is provided on request by myself (corinne.vigouroux@aeronomie.be), or station by station by the individual PIs. I take this opportunity to remind that if any FTIR data is used in a publication, even when the data is released in the public NDACC database, the appropriate PI has to be contacted. If the use of FTIR data is a significant contribution to the publication, a co-authorship should be offered. Otherwise, in agreement with the PI, acknowledgements can be sufficient.

**Minor comments:**

**Please ensure all figures are large in the final version of the manuscript – they are very small and difficult to resolve; it is frustrating. Maybe they could be enlarged by breaking them into separate figures.**

The figures have been enlarged in the AMT version.

**I noticed that in Table 1 the observing period is very long for some of these stations (e.g., Ny-Alesund). Could the authors possibly comment on the instrument stability and performance over such long periods, and if it affects the HCHO retrievals at all? HCHO is difficult to retrieve is not?**

Usually the instrument stability is checked by making regular cell measurements, which allows to verify the good alignment of the instrument (Hase et al. 2009; see also Sect. 2.1 in the present AMTD version). The instruments are re-aligned when a misalignment is detected. Other type of degradation, e.g. of the mirrors, can easily be seen in the decrease in signal to noise ratio (SNR) of the spectra. This decrease in SNR has a direct impact on the precision of HCHO columns (dominant random error source). Therefore, the precision can indeed vary during the time-series period, but the information is anyway provided in the data sets which include random uncertainties associated to individual measurements. The mirrors need to be regularly replaced to avoid a too low SNR. The data sets are also quality controlled after the retrieval process. A too low SNR would lead to a bad root-mean-square (RMS) of the fit, and a threshold on this RMS is used at each station to reject the bad quality spectra. The spectra that do not pass the quality assurance (instrument alignment, RMS,. . .) are removed from the data sets, leading to small gaps in the time-series as seen in Fig. 5.

**Page 7, line 25. The a priori HCHO profile. The approach used here seems sensible, but how sensitive are the retrieval total columns to the a priori – especially as the DOFs is low.**

The effect of the a priori profile (and Svar matrix), is calculated in the smoothing error, which has a random but also a systematic component. The systematic smoothing error component, more closely related to the a priori profile itself, was found to be non-significant in our study (1-2% in most cases, therefore dominated by the other systematic sources, which range from 12 to 26%), as was (too shortly, indeed) written p.11, l.10-11.
This small systematic uncertainty was obtained using the following equation:
(I-A) (xa - <x>) (xa - <x>)$^T$ (I-A)$^T$,
which accounts for the bias of xa, i.e. which accounts for the fact that xa might be different that the expected real <x>, following von Clarmann (2014).
The xa - <x> is obviously not known (otherwise, <x> would be chosen as the correct a priori in the retrievals). Therefore, we had chosen in our AMTD version to use the diagonal elements of the Svar used in Eq. 4 (for the random smoothing error component), as an estimation of xa - <x>.
However, Referee#3 also asked for more discussion about this systematic uncertainty.

We have therefore added the equation above to the new manuscript. We have also decided to use larger values than the ones from Svar for the revised manuscript: we considered the systematic smoothing error that would occur if the a priori profile is differing from the real <x> by -50%, -20%, -10%, +10%, +8%, +5% for the ground-4km; 4-8km; 8-13km; 13-25km;25-40km; 40-120km layers. The values have to vary with altitude to induce a different a priori profile shape: if 50% is used at all altitudes, the a priori profile is then different from <x> by a simple scaling factor, and the systematic smoothing error is close to zero. Using the above values, we obtain systematic smoothing errors from 1 to 9% (median value of 3.4%), which is still small compared to other systematic error sources.

These values rely on the fact that the model WACCM a priori profile shapes are not too far from the reality, which should be the case: due to the known short lifetime of HCHO and its production at or near the surface, we expect that the mean profile peaks at the ground.

This is, as for the random smoothing part, only an estimate of the smoothing systematic error. As discussed in von Clarmann (2014), one would prefer even to not give these smoothing errors at all. We prefer to give them in our paper to provide to the reader as least an idea of the impact of the smoothing in the precision and accuracy of our FTIR HCHO measurements. But these smoothing errors are not provided in the .hdf files that are delivered to the public. When making model or instrument comparisons, the appropriate use of the averaging kernel and a priori profile information (provided in the .hdf files), following Rodgers et al. (2003), allows the user to take implicitly into account the smoothing uncertainty. This means that, for satellite or model comparison, if Rodgers et al. (2003) is used, there cannot be some different systematic biases at different stations due to different xa-<x>.

T von Clarmann, Smoothing error pitfalls, Atmos. Meas. Tech, 7, 3023-3034, 2014.

**Figure 3. The use of atm16 is clearly necessary; HITRAN 2012 needs some corrections...**

Indeed. Spectroscopy is often an issue for atmospheric retrievals. We can only wish that more funding is provided for improved spectroscopic measurements.

**Table 3: The 'DIFF30' is a useful metric, it is given in %, but relative to what? Please be explicit. I'm actually surprised its values are so low (<25%) which is encouraging. Can you also indicate which sites are PROFFIT.**

In the AMTD manuscript, the DIFF30 was given relative to the mean of the daily means for historical reasons (in previous work, the metric used was the standard deviation within a day). This explains why the % values did not correspond to the absolute values divided by the given mean (individual) TC (3rd column of Table 3). However, it is better to give this DIFF30 in percent relative to the mean of individual columns. Therefore, even if the numbers are very close, we have corrected the DIFF30 numbers in Table 3, and have specified in the legend to what they are relative. We have also changed the definition of our mean systematic error (7th column in AMTD): in AMTD version we did: mean(individual Syst / individual TC), and we now do: mean (individual Syst) / mean(individual TC). We then avoid too large effect of outliers or negative small columns.

Indeed, the DIFF30 values, which are, given the lifetime of HCHO of a few hours, an "empirical" measurement of the precision in our FTIR measurements, are quite low (median value of 9%) for a species with such very weak absorptions. We are also very pleased to reach such a good precision. The accuracy is less good (calculated as 14%), and this accuracy should be ideally also empirically evaluated by comparisons with correlative measurements.
The PROFFIT sites were already indicated with an * in Table 3 in AMTD. For AMT, we have changed this by ***, to be more clearly seen.

**Page 13, lines 14-16. Some units are missing**

Corrected for AMT version.

**Page 13, last line. Variability faster than 30 mins. Is there any evidence for this in the literature (e.g., from models, campaigns).**

We did not find any evidence for this tentative explanation from our side. We removed this sentence for AMT version, since it appears indeed too speculative.

**Page 13, line 34: Typo – capital needed at ".this matrix..."**
**Page 15, line 7, Typo. "1E13" should be x1013**

Corrected.

**Figure 5: I can understand why this has been plotted but I think it would also probably be good to show the individual measurements for a single (common) year, rather than over the entire time record at each location. That way you can look at the day to day variability more closely – maybe put such a figure in the supplementary material.**

We followed the Referee's suggestion. Instead of putting this plot of one single year in the supplementary material, we give it in Figure 5 in the new manuscript (chosen common year: 2016; except StDenis: 2011). and the complete time-series are given in supplementary material (Figure S1).

**Figure 6. The variability in the HCHO observations poses some interesting questions. There is a lot of science in here.**

Indeed. We hope that this data set will be soon exploited in modeling studies to explore the reasons for these different observed diurnal cycles. Some discussion and a few comparisons of FTIR and model diurnal cycles have been provided to answer Referee#3 's concerns about the diurnal cycle (but the model diurnal cycles are still not included in the revised manuscript).

**Page 19. The 45% yield reduction – this maybe indeed correct – but can you provide a little more explanation/justification.**

[Figure]

We have added the following sentence in the manuscript: "This fraction of 45% is higher, but of the same order, as the estimated overall impact of deposition on the average HCHO yield from isoprene oxidation (28%), based on IMAGES model calculations. The higher fraction for monoterpenes is intended to account for the impact of the more complex chemistry and larger number of oxygenated intermediates involved in their oxidation, compared to isoprene."

**Page 23: High mountain sites – at such locations it might be wise to quantify the difference between the station elevation and the model elevation for the 2x2.5 degree grid cells (maybe add information to Table 4). Is there any correlation between this difference and the bias?**

Actually, the model takes the altitude of the station into account. We have added the sentence: "The model column is calculated from the calculated formaldehyde profile, between the altitude of the station and the model uppermost level (approximately 20 km), and from the a priori FTIR profile, above that level." Therefore, the overestimation of the model is related to the coarse resolution (2x2.5°), when the mountain site is not in clean area (e.g. Zugspitze, or Altzomoni which is in the same pixel than Mexico City), while the model performs well at mountain sites located in clean areas (e.g. Izaña or Maïdo for which the bias is the same as at StDenis located at the same island but close to sea level).
* * *

---

## Author Comment (AC2) · 25 May 2018

**Reply to Anonymous Referee #3**

**This study presents the retrieval settings of formaldehyde from ground-based FTIR solar spectra, which has been harmonized to allow for consistent retrievals at various stations, under various conditions (remote area, polluted sites, high-altitude sites...). An error budget is presented for each station. The formaldehyde times-series are then presented along with a preliminary investigation of trend and diurnal cycles. Finally, the consistency of the FTIR products is evaluated via comparison with formaldehyde columns simulated by a chemistry-transport model.**
**Developing harmonized formaldehyde retrieval settings through the NDACC (and future affiliated stations) is quite challenging because of the weak absorptions of HCHO in the infrared and the many interferences. This work is therefore valuable in the framework of validation efforts of model simulations and of current and future satellite instruments. The topic developed here fits perfectly the scope of AMT. The paper is globally well written and the structure is clear. Nonetheless, some results/figures are not adequately presented, which somehow impedes a proper evaluation of the results (see major comments here below). Therefore, I recommend publication of this study after addressing the comments listed hereafter.**

We thank the referee for his/her careful review and his/her constructive comments. We have replied below and changed the manuscript accordingly for the AMT version.

**Major comments:**

**Additional effort is needed to present the results more synthetically and to make some figures easier to read. Fig. 5-8 are particularly difficult to read because of the numerous small panels. It is really unfortunate because these figures present the main results of the study. I assume that the large number of subplots makes them difficult to display, but I really think that such figures deserve a reshape.**

We definitively agree that these figures are difficult to read in the AMTD version. We have enlarged them for the AMT version.

**In particular:**

**- The various x-axes in Fig. 5 and 7 don't help the reader. Some seasonal cycles appear completely squeezed because these panels encompass >15 years, while others represent 2-3 years only on a panel of the same size. Please display the time series on an x-axis common to all the stations.**

For Fig. 5: This point has been also raised by Referee #1. Following his/her suggestion, we have therefore chosen to show a plot with a common year to all stations (2016), except for St-Denis (for which, due to lack of measurements in 2016, we have chosen 2011). The complete time-series as in Fig. 5 will be given in supplemental material (Fig. S1).

For Fig. 7: the x-axes are already common for all stations in the AMTD version (Jan 2003 – Dec 2016). It is Jul 2002 – Jul 2017 in the AMT version.

**For Fig. 5 and 7, I also suggest to gather some time-series within the same panels, e.g., following the subdivision in the text (i.e. clean, intermediate, polluted sites), using different colour lines. The interest to gather time series within the same panels would also be to help the reader to appraise the large panel of HCHO columns covered by the FTIR sites.**

We understand the point of view of the referee, and we had hesitated to use a such division in the AMTD version. But, due to Fig. 8 and the discussion with the model comparisons (that mainly follows location/latitude because location is the main reason for different model behaviors), and to increase the facility to a reader only interested in one station to find it at the same place in all Figures and in all Tables, we have decided to keep a latitudinal order.

However, we find the suggestion of the referee to use different colors very helpful to stress the different concentration levels. Therefore, we use three different colors in the new Fig. 5 to 8. (blue, orange and red, for clean, intermediate and polluted sites).

**- For Fig. 6, perhaps it is not needed to display all the diurnal cycles in the manuscript. A solution would be to keep here only a few representative of those to support the discussion. The others can be moved to supplementary material.**

We followed the Referee's suggestion. Figure 6 gives the diurnal cycles for only 10 of the stations in the new manuscript, and the other ones are given in the supplementary material (Fig. S2).

**In Fig. 7-8, it is very hard to distinguish the raw model data from the smoothed ones.**

We have enlarged Fig. 7 for a better visibility. However, the main reason why it is hard to distinguish the raw model data to the smoothed one, it that in general the effect of the smoothing is rather small. This is in line with the small reported smoothing uncertainty, and with the fact that the FTIR a priori profile shapes (from the model WACCM) are close to the IMAGES profile shapes.

**I also find the discussion on the basis of Table 4 quite "raw". The authors made huge efforts to harmonize the retrievals and to produce a consistent pattern of HCHO measurements worldwide. There could be, along with Table 4, a map including in colour background the mean HCHO from the model over 2003-2016, and the mean FTIR HCHO in dots filled following to the same colour bar, at the location of each station. In one glance, the reader would have a good overview of the pattern of FTIR measurements as well as of the overall consistency with the model.**

Indeed, the discussion of the model comparison is quite "raw" in this paper which aims primarily at presenting the harmonized retrieval strategy and the overall network of HCHO FTIR data sets. The model is only used to show the consistency in terms of bias / seasonal cycles. That is also why

this paper is published in the AMT journal. More discussion on comparisons with models will be the subject of future publications.

However, this is indeed a good suggestion to provide such a map. We prefer to provide it as supplementary material (as Fig. S3; IMAGES climatology for 2005-2015, values in $10^{15}$ molec/cm$^2$), and we give it also below for discussion (Fig. 1), because the FTIR and model data cannot be quantitatively compared with such a map: the model data are plotted for the 2005-2015 period, while the FTIR data usually cover different time-period (e.g. Porto Velho has only values in 2016-2017). Furthermore, in such a map the model is calculated for the model surface of course, while the mountain sites have a high elevation altitude. For this reason, we prefer to show high altitude sites (>2km) as black crosses. The map shows indeed very clearly the different levels of HCHO covered by the FTIR stations. The agreement with the model looks good (very clean Arctic sites, intermediate European sites, large gradient levels in the Southern Hemisphere from the clean site Lauder, then StDenis, followed by the higher levels at Wollongong, and the strong maximum in Porto Velho). However, this comparison can unfortunately only be qualitative, because of the differences in model and data temporal sampling used for this map.

[Figure]

Figure 1: Climatological daytime HCHO columns (2005-2015, 8-17 h local time) calculated by the IMAGES model (in $10^{15}$ molec/cm$^2$). The long-term averaged HCHO columns at the FTIR stations are shown as filled circles using the same color code. The high-altitude stations (for which the comparison with the model is severely biased due to surface altitude difference) are denoted by crosses.

**A single HCHO a priori profile is used at each station for the retrievals. This assumes that not only the shape, but also the HCHO concentration simulated by the model, are quite reliable. What is the impact of another HCHO a priori profile on the retrieved columns? e.g., an a priori from another model that would be significantly different, or again an a priori that would be derived from other measurements (like ACE-FTS)? For a very weak absorber like HCHO, with little retrieved vertical information, I expect the impact to be, if not substantial, at least not negligible. For example, it is clear from the very similar shapes of the a priori and retrieved profiles in Fig. 4, that the retrievals are dependent on the a priori. It is important to discuss this point and to add this component to the error budget.**

Indeed, with 1 to 1.5 DOFS only, the retrieved profiles follow the shape of the a priori profiles. Please, see the detailed reply to Referee #1 about the effect of the a priori profile.
We did not use another model or ACE-FTS data to evaluate the possible difference in a priori, because of the high number of stations there, and the relatively small impact of this smoothing systematic uncertainty. In addition, ACE is not providing profiles down to the ground where the largest difference is expected. Instead, we have chosen a common bias in a priori profiles for all the stations (see reply to Referee #1).
We have re-evaluated it from 1-2% (when $S_{var}$ was used in the AMTD version) to a median value of 3.4%. We have added an equation and some additional text in the new manuscript, and 2 columns in Table 3 (systematic smoothing error and total systematic error).

**The error budget is established for each station on the basis of a single measurement. Why one measurement only per station? I find this very reductive, especially that it is not even said whether this single measurement is representative for the whole data set (DOFS, residuals, total column...). Hence, one can easily casts doubts about such error budget. This should be made ideally with a representative subset of FTIR measurements, covering different seasons, different zenith angle, etc.**

Sorry if this was not clear in the AMTD version, but the error budget is made for each single measurement, and the mean of all individual errors is reported on Table 3. We think that the misunderstanding comes from the sentence p. 11, l.11-15 (and from the legend of Table 3). When we wrote "… on one individual HCHO FTIR total column measurement.", we meant that the reported uncertainties are valid for a single measurement (and not e.g. for the monthly means that are used for model comparisons). In l.12, it is written "…the mean of the random uncertainty…". But we agree that this is not clear enough, so we have modified the sentence and the legend.

**Page 14, Lines 9-16: Since you know that models usually underestimate the natural variability of HCHO, and since you know the impact of such underestimation on the smoothing error estimation, why wouldn't you increase the variability of the model to be more representative for the real variability? Knowing the difficulty for the models to simulate highly-variable reactive gases like HCHO, a model variability multiplied by e.g., 2, would still be conservative.**

The estimation of the smoothing error is a delicate subject. It is supposed to give meaningful values when a real variability matrix Sa is available, which is usually not the case. For this reason, the FTIR data sets that are archived in the NDACC database (for official target species such as ozone, CO, CH4,..) do not include the smoothing uncertainty. The user is asked, if he needs this information, to find by himself the most accurate information available at the time he is using the data (which may not be the same as at the time the data sets are archived in NDACC, if more / better correlative measurements / modeling studies are made available in between). Then he can calculate the smoothing error according the AK provided in the files, and using Eq. 4 of our manuscript.

Another reason for not providing the smoothing error is the fact that if the data are used for validation, then with the appropriate use of the FTIR AK prescribed in Rodgers and Connor (2003), this smoothing uncertainty component is discarded.

Here, we decided to give an estimation of the smoothing error based on our present knowledge of model calculations (here WACCM). This smoothing error can be improved in the future, when model variability will be improved or when more correlative profile measurements will become available. The Sa matrix is currently calculated from several decades of model output, and the dominant variability (also seen in the measurements) come from the seasonal cycles. If the IMAGES model shows indeed usually less variability than the FTIR measurements (Table 4, last column), it seems that it is worse in some cases (Mauna Loa, Izaña) than in other ones (Ny-Alesund, Toronto). So it would seem arbitrary to choose to multiply by e.g. 2 at all stations.

We therefore prefer to keep our estimation as it is in the AMTD version, knowing that it is not perfect.

The example given for Saint-Denis (comparing Vigouroux et al., 2009, where aircraft measurements were used for constructing the Sa matrix, and the present work) looks maybe too extreme (from 14% to 2% for the respective smoothing errors). The Sa matrix used in Vigouroux et al. 2009 had as large values as 70%, which is probably overestimated given the low FTIR variability observed there. We have added a few words in the manuscript about this.

However, if a reader would like to know the smoothing error that would be due to a doubled variability Svar matrix (i.e., all – diagonal and non-diagonal – elements multiplied by $2^2=4$) the result is straight-forward: the smoothing uncertainty would be doubled.

**Page 15, Lines 15-27: The diurnal cycle is sometimes very weak. Furthermore, the midday observations (low zenith angle) probing less atmosphere, we can expect larger uncertainties associated with such measurements. Hence, owing to these larger midday uncertainties, are the diurnal cycles still significant? Or couldn't some of these diurnal cycles (e.g., the midday minimum found at some sites) be just the effect of larger uncertainties and less sensitivity associated with the low zenith angle?**

The averaging kernels (AK) for the midday measurements are similar to the morning and afternoon ones, especially below 15km, where most of the HCHO lies. Only the very high solar zenith angles

give (at some of the stations), significant enhanced degrees of freedom for signal (DOFS), and this enhanced sensitivity is not located at the altitude where most of the HCHO lies. To illustrate this, we give below a plot of the AK (Fig. 2), for a station with a minimum diurnal cycle observed at midday (Sodankyla) and with a station where a maximum is found at midday (StDenis), for local time (LT) = 12 (left panels), and local time at high solar zenith angle, LT=19 and 17 for Sodankyla and StDenis, respectively (right panels). The DOFS (therefore the sensitivity) is very similar in the ground-15 partial column, and very close to 1 at both stations also at midday (0.98 and 0.96 at Sodankyla and StDenis, respectively). We do not show the plots for e.g. a local time of 15, because they are actually the same as at midday, and the DOFS (total and for the ground-15km partial columns) are exactly the same (0.98 and 0.96 at Sodankyla and StDenis, respectively).

[Figure]

Figure 2: Averaging kernels at Sodankyla (upper panels) and StDenis (lower panels) at Local Time (LT)=12 (left panels) and 17 (right panels). The DOFS for total columns and partial column (ground-15km) are also provided.

Also, the uncertainty budget is not significantly larger for the midday measurements.

Furthermore, the diurnal cycles are really different from station to station, with sometimes indeed a midday minimum, but sometimes a midday maximum, and sometimes a continuous increase from the morning to the late afternoon. Since the technique and the retrievals settings are harmonized among the stations, the Referee's suggested dependence on zenith angle should be reflected in all stations, which is not the case.

We therefore think that the observed diurnal cycles are true.

Maybe, some references to literature is missing in our AMTD version to strengthen our results. The problem is the sparse data providing diurnal cycles, and often at different locations than our stations. And as we just discussed above, the diurnal cycles seem very site dependent. However, in e.g. De Smedt et al. (2015), diurnal cycles from a few MAXDOAS stations show indeed various behaviors: very weak diurnal cycle at OHP (Southern France) in Winter and Spring; wide minimum around midday at Beijing and Xianghe in Spring and Autumn, and constant increase in Summer (as observed for Bremen, Toronto and Paris). This reference to MAX-DOAS observations have been added in the revised manuscript.

We found a paper providing diurnal cycles at Zugspitze (Leuchner et al., 2016). This study is using surface measurements so is not fully comparable to our measurements (therefore, we did not add this reference in the revised manuscript). A rather weak diurnal cycle is found in winter as in our FTIR measurements, but in Spring and Autumn, a maximum is found around midday, in opposition of our minima in these seasons. In summer, a larger diurnal cycle is found, more centered in the afternoon which is also observed from FTIR measurements.

The diurnal cycles observed at the close station Jungfraujoch by FTIR measurements (Franco et al., 2016) are showing, for all months of the year, a midday maximum, which is very different from the Zugspitze FTIR diurnal cycles. The IMAGES model shows diurnal cycles in phase agreement with our FTIR measurements at Zugspitze except for the Summer (Fig. 3 below, upper panels). We also give the FTIR and IMAGES diurnal cycles at Sodankyla, StDenis and Maïdo to illustrate that the model also show different diurnal cycles at different locations and seasons, even very close ones (St-Denis/Maïdo). In some cases, a minimum is indeed calculated at midday, in StDenis the maximum is around midday – 1pm. The model also reproduces quite well more variable diurnal cycles as in Maïdo. We have added the Jungfraujoch diurnal cycles in the discussion, suggesting that more investigation is needed to understand the observed different diurnal cycles.

We did not include the model diurnal cycles in the AMTD paper, because it is a paper focusing on the harmonization of the FTIR data, and there is so much science that can be exploited from these data sets that it deserves several separate papers.

Note also, that the model is not always in agreement with the FTIR diurnal cycles, e.g. the strong maximum at Mexico City is not reproduced., the model providing very weak cycles there (not shown).

[Figure]

Figure 3: FTIR and model IMAGES diurnal cycles at four stations, for the four seasons.

**Section 3.3: The investigation of the trends are very preliminary, and there is no discussion of the results. If, as quoted in this section, a more comprehensive investigation is beyond the scope, I don't see the interest of this section as it is currently. Or I recommend to add a bit of discussion, e.g., how do significant trends compare to other trends in the literature from, e.g.: De Smedt et al. (2010), De Smedt et al. (2015), Franco et al. (2016), Jones et al. (2009).**

We thank the referee for the interesting suggestion. We have added such a discussion in the AMT version. The signs of the FTIR observed trends look indeed in good agreement with the previous studies De Smedt et al. (2015) and Franco et al. (2016).
We did not compare with Jones et al. (2009) because the period of concern in this study (1992-2005) is too different from ours (2001-2016). Since the De Smedt et al. (2015) study is going further than De Smedt et al. (2010), we only compare to the last version of the satellite work.

**There have already been comparisons between previous HCHO columns from the FTIR and from UV-Vis instruments (satellites and MAX-DOAS), showing in overall a good agreement. However, it is obvious from this study that the HCHO retrievals from the FTIR are very sensitive to the retrieval choices (spectroscopic database, micro-windows, a priori...). Biases up to 50 % are even mentioned between different retrieval approaches. This means that the harmonized retrievals presented here can potentially improve or deteriorate significantly the comparisons with the UV-Vis instruments. I think this point needs to be discussed, or at least mentioned in the conclusion.**

Concerning the 3 studies where FTIR have been compared to MAX-DOAS and/or satellite instruments:

- At Reunion Island (Vigouroux et al., 2009): we have compared both FTIR data sets (this "new" study and the "old" 2009 data set), and the bias is only of -1.4% (new-old / old). The comparisons FTIR-MAX-DOAS showed a bias of -8.4%, so the comparisons with the new FTIR data would still be within the systematic error budget on the differences (10%).

- At Lauder: there is indeed a high bias (-49%) between new and old data sets (new-old / old). The new data set would then be in worse agreement with the GOME data set used in Jones et al. (2009), where FTIR data showed a small low bias compared to the satellite. But this needs to be re-evaluated with the reprocessed QA4ECV satellite data (De Smedt et al., 2018), because the satellite products have changed a lot.

  Furthermore, the new data set is in much closer agreement with the simulation of four different models that were all of them 50% lower than the old Lauder and Wollongong data sets (Zeng et al., 2015). The Wollongong and Lauder data in Zeng et al. (2015) were using the same retrieval strategies (described in Jones et al., 2009). This high bias between the two strategies is very likely due to the 2869.65-2870.0 cm$^{-1}$ window used in Jones et al. (2009).

- At Jungfraujoch: the strategy used in Franco et al. (2015) does not include the problematic window of Jones et al. (2009). Therefore, we do not expect such a large bias depending on strategies. It requires some time to make new analysis with different strategies (reason why Jungfraujoch is not included in the present study), so this cannot be evaluated for the present review. What we have done up to now to evaluate the impact of the settings is to run our retrievals at Maïdo with the micro-windows (mws) and the spectroscopy used in Franco et al. (2015). If both mws and spectroscopy are changed, then the bias is only of -4% (new-old / old). If the same bias is assumed at Jungfraujoch (which is really "simplified" since the 2 sites have different atmospheric conditions), then the FTIR data would go closer to the model (-8% instead of -12%), therefore in closer agreement as well with MAX-DOAS. However, if only the mws are changed (i.e. if we use the mws of Franco et al. 2015, and the spectroscopy atm16), then a bias of -25% is observed at Maïdo. The small bias of 4% is therefore due to a compensation of opposite effects of mws and spectroscopy. This illustrates again why this is so important to build such harmonized

network where the parameters inducing systematic uncertainty sources are consistent, removing internal bias within the network.

We will add some discussion about this in the AMT version, for Reunion Island and Lauder/Wollongong. We will not do it for Jungfraujoch since the few tests mentioned above was made in Reunion Island atmospheric conditions, so the results are not directly applicable to Jungfraujoch. Once Jungfraujoch will join this harmonization effort, new comparisons with MAX-DOAS will be able to confirm (or not) the discussion above.

Zeng, G., Williams, J. E., Fisher, J. A., Emmons, L. K., Jones, N. B., Morgenstern, O., Robinson, J., Smale, D., Paton-Walsh, C., and Griffith, D. W. T.: Multi-model simulation of CO and HCHO in the Southern Hemisphere: comparison with observations and impact of biogenic emissions, Atmos. Chem. Phys., 15, 7217-7245, https://doi.org/10.5194/acp-15-7217-2015, 2015.

**Minor comments**

**Page 1, Line 3: "accurate and precise". In light of the error budget and the large biases depending on the retrieval choices ("as large as 50%", Line 5), this statement should be dampened.**

This is still a good accuracy when compared to satellite products, and with improved spectroscopy we believe that the accuracy is much better than 50%. We found a median systematic uncertainty of 13% and a maximum of 27%. However, we followed the referee's suggestion since in the past studies this might be not true. Therefore, we changed to "several independent studies have shown that the FTIR measurements can provide formaldehyde total columns with a good precision".

**Page 1, Line 8: stations. Most of them**
Done.

**Page 1, Line 11: Change ";" to ","**
Done.

**Page 2, Line 1: Unclear. Is it for the systematic or the random uncertainties?**

Changed: "Depending on the station, the systematic and random uncertainties of an individual HCHO total column measurement lie between 12 and 27\%, and between 1 and 11$\times 10^{14}$ molec/cm$^2$, respectively. The median values among all stations are 13\% and 2.9$\times 10^{14}$ molec/cm$^2$, for the systematic and random uncertainties, respectively."

**Page 2, Line 11: NOx is not defined yet**
Done.

**Page 2, Line 16: of only a few hours**
Done.

**Page 2, Line 17: and to test**
Done.

**Page 2, Line 32: at a few locations**
Done.

**Page 3, Line 3: Change "geographical" by "spatial"**

We prefer to keep "geographical", because the variability is more related to "geography" (type of land; ocean;…) rather than on spatial distance.

**Page 3, Line 3: A lot of efforts are**
Native English colleague advises us either "A lot of effort is" or "Lots of effort are". Therefore, we kept the sentence as it is in AMTD version.

**Page 3, Line 7: stations that will also be part**
Done.

**Page 3, Lines 13-16: Isn't it because HCHO is so challenging to retrieve that it is not (yet?) a standard product from the NDACC FTIR?**

Indeed, the two reasons are somehow linked. But official NDACC targets also include some challenging species, such as HCN and C2H6 (small absorptions) or CH4 (spectroscopic issues). Historical choices have been made in favor of some gases among others.

**Page 3, Line 27: monthly mean time-series**
Done.

**Page 4: Fig. 1 would deserve to be a bit larger due to e.g., the concentration of stations in Europe**
Done.

**Page 4, Line 14: (1995) and/or Hase et**
Done.

**Page 4, Line 15: pressure- and temperature-dependent**
Done.

**Page 5, Lines 5-6: It is said elsewhere in the manuscript that the use of different retrieval parameters can substantially affect the retrieved columns. In particular, the use of either HITRAN 2004, 2008 or 2012 for the HCHO spectroscopic parameters leads to very large differences in the retrieved columns. I would have expected the authors to better motivate their choice of HITRAN 2012, especially that eventually some lines of interfering species needed to be empirically adjusted in this spectroscopic database.**

HITRAN 2012 has been chosen because it includes the latest improved HCHO parameters (broadening coefficients, Jacquemart et al., 2010), which complements the release in HITRAN

2008 of new line intensities from the same group (Perrin et al., 2009). We will add this information, and the references, in the manuscript.

The motivation is not coming from the interfering species. Especially, this is true that for some interfering species, the oldest HITRAN versions can be better than HITRAN 2012 (e.g. for CH4). By the way, the empirical adjustment made in atm16 is, in some cases, simply to use an oldest database (e.g. for H2O, or, in the 2781 cm$^{-1}$ window's case, CH4). The use of atm16 ensures us that for each species the best spectroscopy is used. The work is then done by one of us (G. Toon) and the IRWG community can make use of it, without redoing all the databases comparisons and/or adjustments. Furthermore, this spectroscopy is publicly available (the link is provided in the manuscript), so the users can have easily access to it.

For the Fig.3, it was chosen to plot our spectroscopy atm16 in comparison with HITRAN 2012, in order to stress some remaining problems in HITRAN 2012 for the interfering gases. This can be useful for the spectroscopic community which is often not aware of such specific problems.

**Page 6, Line 3: that is distributed**
Done.

**Page 7, Line 8: Rodgers (2000)), is**
Done.
**Page 7, Lines 6-9: Is the little gain in information the only reason why you keep these two windows?**

No: the sentence says "which contain less absorption for interfering gases", which is an advantage of these 2 micro-windows. But the gain in information is small, therefore if one decides not to use them, there would be little difference in the retrievals.

**Page 7, Lines 10-11: Is this individual spectrum representative for the whole time-series? How do its DOFS and its fitting residuals compare to the other observations?**

This individual spectrum has been chosen because it is typical for Maïdo. It corresponds to a column of $25x10^{14}$ molec/cm$^2$, while the mean is $20x10^{14}$ molec/cm$^2$. The DOFS for this spectrum is 1.1 (mean of 1.2), and the root-mean-squares (RMS) of residuals is 0.11 (mean of 0.12).

We will add the information on the DOFS and RMS of this specific spectrum in the manuscript.

**Page 7, Line 26: WACCM v4 (Garcia et al., 2007).**
The reference is for WACCM v3 (we are not aware of a reference for v4). We propose to change with "the v4 of WACCM (Garcia et al., 2007)".

**Page 7, Line 29: not to use**
Done.

**Page 7, Line 33: Sussmann et al. (2011), and**
Done.

**Page 8, Fig. 2 caption: total column of**
**(the same in Fig. 3 caption)**

**And further, same line: Change "The figures in the lower panel are" to "The lower panels are**
Done.

**Page 10, Line 31: DOFS, in Table 3**
**Same line: provide more than**
Done.

**Page 10, Line 33: (upper panels)**
Done.

**Page 11, Fig. 4 caption: Upper panels: averaging**
**And further: Lower panels:**
Done.

**Page 11, Line 5: associated with**
**Page 11, Line 6: (lower panels)**
Done.

**Page 11, Fig. 4: Do you have an explanation for the contribution from the high-altitude averaging kernels (in green)? It looks a bit odd to have a contribution from such layers to the HCHO retrievals. Still Fig. 4: From the shape and the high values of the total column averaging kernel, don't you "overfit" the HCHO retrievals?**

The shape of the total column averaging kernel cannot come from an overfitting of HCHO: Kiruna (and other stations who only scaled HCHO) obtains a similar behavior: smaller than 1 below 3-5 km, and larger than 2 above 10-15km. This is also not due to our specific retrieval strategy since this behavior is seen in Jones et al. (2009) and Viatte et al. (2014) as well. In Franco et al (2015) the total column averaging kernel (AK) is not provided.
Therefore, we believe that this is due to the specific spectroscopic parameters and the atmospheric information than can be obtained for the spectra (the localization of the altitude where the variability takes place cannot be well defined, and an underestimation of the variability below 5km is compensated by an overestimation above).
On the other hand, concerning the green high-altitude AKs: these ones appear significant only at stations where the DOFS are larger than 1. In these upper panels, the mean AKs are provided, and if we look at individual retrievals the green ones appear (or are larger) in the case of high solar zenith angles (see Figures above, provided in the diurnal cycle discussion). It is not straightforward to verify if these are real or due to overfitting. The regularization strength has been let to the appreciation of the PIs, using the L-curve method, knowing that the measurement noise and local conditions are site dependent. This is not a very strict method, especially for weak absorption gases such as HCHO, so it might indeed be that the highest DOFS (1.4-1.6) are overestimated. However, the regularization strength (in the reasonable range as provided in our study) has little influence on the retrieved total columns, as tested at Maïdo using a scaling of HCHO: only 2% of bias was observed compared to the present DOFS of 1.2.

We note here that, at the very high solar zenith angles the AKs at Porto Velho were found to have unrealistic values (strong negative oscillations), which was not detected in the AMTD version, since the AKs were good for low to medium/high solar zenith angles. The final retrieval data sets have then be more constrained, leading to a mean DOFS of 1.1 instead of 1.3, with a mean bias effect on total columns of only 1.3\%.

**Page 12, Table 3, as well as in the manuscript: Providing the random uncertainties in total column only is really misleading, especially in the discussions. Each time, it forces the reader to look at Table 3 and to calculate the percentage before knowing whether it is significant for the station that is considered. I recommend to provide all the uncertainties (also) in percentage of the total column (you already do it for the systematic uncertainties).**

The random uncertainties have been provided in absolute values, since this is the relevant quantity in terms of precision and detection limit. (e.g. satellite requirements for ground-based validation are given in absolute values). Also, similar values are expected among the stations in absolute units, and a percentage value would make a station with clean levels of HCHO appear less precise. The systematic uncertainties are given in percentage because they are expected to be similar among the stations in percentage (not in absolute values).
Therefore, we prefer to keep the absolute values in the discussion for random uncertainties as for the DIFF30 which are calculated to evaluate these random errors. In principle there is no need to make the conversion of random errors in % to follow the discussion. However, since previous studies provided their random error budget in %, we will follow the suggestion of the referee and add for easier comparison the % values for the Total Random error. For the detailed Random and Smoothing errors in %, the reader can simply divide by the mean TC given in the 3$^{rd}$ column.

**Page 13, Lines 25-29: Why such exceptions for these SFIT4 stations, and not for others? Isn't the error budget supposed to be fully harmonized among all the SFIT4 stations?**

The formalism for error calculation (Rodgers 2000) is harmonized. But, we could have some underestimation/overestimation at some stations due to the Sb matrices, e.g. the Sb matrix for temperature might be realistic at one station and optimistic/pessimistic at another one. Also, the Svar matrix used for the smoothing random error has been taken from the WACCM variability at each station, and the model might provide more realistic variability for some stations than for other ones (as IMAGES does). We see for Table 3 that the smoothing error can contribute significantly to the total random error in some cases (St-Petersburg, Toronto, Boulder, Porto Velho,…) while it has little impact at Lauder and Ny-Alesund (from WACCM).

**Page 13, Line 29: might have**
**Page 13, Line 33: stratosphere. This matrix**
**Page 13, Line 34: while for the PROFFIT users, these values**
**Page 14, Lines 3-4: rephrase as "considering the random uncertainty in Table 3 (4$^{th}$ column) is sufficient" to avoid misleading**
**Page 14, Line 24: HCHO line intensity.**
**Page 14, Line 26: the PROFFIT channelling source (from 7 to 17 %), which also has a systematic component. We see from Table 3**
**Page 15, Line 5: few 1 x 1013 molec/cm2**

**Page 15, Lines 15-16: Bad sentence. "To reconcile the different... (afternoon), it is crucial to have ground-based...**
**Page 15, Line 21: mid-latitude cities**
All done.

**Page 18, Lines 1 and 5: The use of "time step" is here misleading with the computational time step of the model. I suggest to use "with outputs every 6 hours/ 20 minutes" instead.**

We keep "time step" because we indeed mean the computational time step of the model. The outputs are daily averages. This last information is included in the new manuscript.

**Page 18, Line 11: delete one "the"**
**Page 18, Line 22: in Bauwens et al. (2016)**
**Page 19, Line 1: justified by the**
All done.

**Page 19, Lines 3-6: How do you deal with the model surface that is below the altitude of the station (which should be the case for most of the mountain sites), especially where there is a substantial altitude difference?**

The IMAGES model provides profiles (not total columns), so the columns from the model are calculated from the profiles starting at the altitude of the station (removing the profiles levels between the surface and the altitude of the station). A sentence has been added in the manuscript.

**Page 19, Lines 7-9: Do you mean that you re-scale the model outputs at the time of the FTIR measurements? Or do you use the nearest model output to each FTIR data?**

The global model output is given once per day. But an offline calculation of the diurnal cycle (at each model pixel) is used to re-scale the model data at the time (hour) of each FTIR measurement. In the first paragraph of Section 4.1, we replaced the sentence "The effect of diurnal variations is accounted for..." by the more complete description: "The model calculates daily averaged concentrations of chemical compounds. The effect of diurnal variations is accounted for through correction factors on the photolysis and kinetic rates obtained from a full diurnal cycle simulation using a time step of 20 minutes. The same model simulation also stores on files the diurnal shapes of formaldehyde columns required for the comparison with FTIR data."

The last sentence of Section 4.1 is replaced by: "Also, the local time of each observation is taken into account by re-scaling the daily averaged concentration using the formaldehyde diurnal shape factors calculated by the model with a time step of 20 minutes."

**Page 19, Line 24: "within a month" is redundant**
**Page 19, Line 25: "The median of IMAGES and FTIR differences" or "The median of IMAGES and FTIR biases"**
**Page 19, Line 30: change ";" to ","**
Done.

**Page 21, Fig. 8 caption: "in coincidence with". Do you mean the same day, within 20 minutes of each FTIR data?**

Yes, the same day. Then, the daily output of the model is scaled at the time (hour) of the FTIR measurement using the diurnal cycle calculated independently using a time set of 20 minutes. (so NOT within 20 minutes of each FTIR data, but same hour).

**Page 23, Line 13: Is Boulder (~1600 m asl) really a urban site?**

Boulder is a city of about 100 000 habitants. As for Wollongong, the emission sources should be both from biogenic and anthropogenic origin. We will keep Boulder in the "Mid-latitude cities" section, but remove the term "urban" in l.13.

**Page 23, Lines 29-30: At such a remote site, the dominant source of HCHO should be CH4 oxidation. I don't think that other sources from continental areas can be significantly at play here.**

The referee is right that this is too speculative. We will change the sentence by "The reasons of the pronounced observed variability are unclear at present."

**Page 24, Lines 5-9: Could it be also due to the FTIR technique, which measures in clear-sky conditions only? The FTIR would sample only air masses free of huge emissions and hence would underestimate the gas abundance in this region.**

We don't believe that the bias is due to FTIR measurements sampling. The sampling of the measurements is actually very high during the biomass burning season due to the low cloudiness during the dry season. Furthermore, the model overestimates the FTIR measurements also during the background season, as seen as well as at Paramaribo. As written in the AMTD version, the overestimation of IMAGES over Amazonia was already found in Bauwens et el. (2016).

**Page 25, Lines 14-15: We do not aimed at evaluating the model, but at showing that the FTIR**
Done.

**I have here an open question, which I think is relevant for a data set that is designed to be used for intense model and satellite validation efforts. Will this data set be made publicly available? And if yes, will there be an effort to fully harmonize the archives, the file format, and the way the data are saved? There are currently inconsistencies between FTIR data sets from different stations (especially for the AKs). Such inconsistencies sometimes refrain external users to use NDACC FTIR data, while such data sets deserve to be easily accessible and as user-friendly as possible for non-community users.**

Concerning this data set to be made publicly available: as also replied to Referee #1, this is currently under discussion within the InfraRed Working Group (IRWG) community, and a final decision will be taken at the next IRWG meeting in June. The data will be very likely downloaded

in the public NDACC repository. In the meantime, the data are provided on request by myself (corinne.vigouroux@aeronomie.be).

Concerning harmonization of the archive: we believe that these last few years, lots of effort have been made to harmonize the file format (geoms hdf files). A few inconsistencies remain, e.g. descending or ascending vertical grid, profiles given in the middle of the layers or at the altitude levels. This is quite easy to be solved by the users, but however, it is certainly important to improve this as soon as possible. Furthermore, it is possible that a few stations are still not in line with the current IRWG format requirements. This is indeed crucial for the NDACC database to receive some feed-back from the users, when the inconsistencies still persist. This question will be raised at the next IRWG meeting. We encourage the users to contact the chairmen for reporting inconsistencies: https://www2.acom.ucar.edu/irwg/contacts.

---

## Referee Report (RR1)

Review of the revised version of "*NDACC harmonized formaldehyde time-series from 21 FTIR stations covering a wide range of column abundances*" (amt-2018-22)
by *Vigouroux et al.*

The authors did a great job addressing my comments and concerns, and adequately revised the manuscript. It has been improved by the addition of technical explanations that better motivate their choices related to the retrieval setup. The new formaldehyde products are also more discussed, especially in light of previous studies. Improvements were also brought to the figures to make them and the corresponding discussion easier to follow. The current manuscript fits the scope and standards of AMT. Therefore, I recommend to accept this revised manuscript for publication.

Here are just a few typos I spotted in the text:

- P. 15, line 23: Empty space missing in "and a priori"
- P. 16, line 11: "Indeed, we can…"
- P. 19, line 2: "Rondônia"
- P. 22, line 33: Empty space missing in "(+6 / -3%)"
- The same P. 24, line 8
- P. 27, line13: delete "

I would like to apologize for the delay before reviewing this revised manuscript. Because of a wrong email address used by Copernicus, the initial call for review has not been delivered to me.